# SSOLE: Rethinking Orthogonal Low-rank Embedding for Self-Supervised Learning

**Lun Huang**[1,2]    **Qiang Qiu**[3]    **Guillermo Sapiro**[2,4]
[1]Duke University    [2]Princeton University    [3]Purdue University    [4]Apple
lun.huang@duke.edu    qqiu@purdue.edu    guillermos@princeton.edu

## Abstract

Self-supervised learning (SSL) aims to learn meaningful representations from unlabeled data. Orthogonal Low-rank Embedding (OLE) shows promise for SSL by enhancing intra-class similarity in a low-rank subspace and promoting inter-class dissimilarity in a high-rank subspace, making it particularly suitable for multi-view learning tasks. However, directly applying OLE to SSL poses significant challenges: (1) the virtually infinite number of "classes" in SSL makes achieving the OLE objective impractical, leading to representational collapse; and (2) low-rank constraints may fail to distinguish between positively and negatively correlated features, further undermining learning. To address these issues, we propose **SSOLE** (Self-Supervised Orthogonal Low-rank Embedding), a novel framework that integrates OLE principles into SSL by (1) decoupling the low-rank and high-rank enforcement to align with SSL objectives; and (2) applying low-rank constraints to feature deviations from their mean, ensuring better alignment of positive pairs by accounting for the signs of cosine similarities. Our theoretical analysis and empirical results demonstrate that these adaptations are crucial to SSOLE's effectiveness. Moreover, SSOLE achieves competitive performance across SSL benchmarks without relying on large batch sizes, memory banks, or dual-encoder architectures, making it an efficient and scalable solution for self-supervised tasks. Code is available at https://github.com/husthuaan/ssole.

## 1 Introduction

Self-supervised learning (SSL) (Chen et al., 2020a; He et al., 2020; Bardes et al., 2022) learns meaningful representations from unlabeled data by exploiting the intrinsic structure within the data, reducing the dependence on costly labeled datasets. SSL has become crucial in fields like computer vision, natural language processing, and speech recognition, enabling models to harness vast amounts of unannotated data.

Orthogonal Low-rank Embedding (OLE) (Qiu & Sapiro, 2015; Lezama et al., 2018), originally developed for supervised image classification, constructs a feature space where same-class samples reside in low-rank subspaces, enhancing intra-class similarity, while different-class samples are orthogonal, promoting inter-class dissimilarity. OLE operates at the matrix level, optimizing the nuclear norm as a surrogate for rank, making it well suited for leveraging multiple views or augmentations of data in SSL. Integrating OLE into SSL offers several benefits: (1) Its matrix-level operation is well suited for leveraging multiple views or augmentations inherent in SSL; (2) Enforcing low-rank constraints on positive pairs not only brings their representations closer but also minimizes the number of factors controlling the image representation, leading to more compact features; (3) Enforcing high-rank constraints on negative pairs not only pushes their representations apart, but also prevents dimensional collapse, maximizing the representational capacity of the embedding space.

However, OLE has not been successfully applied as a standalone metric in SSL. Previous works have only partially utilized OLE's components. For example, LORAC (Wang et al., 2022) incorporates low-rank embedding as a regularization term within the MoCo framework (He et al., 2020) but relies heavily on contrastive loss. Methods like VICReg (Bardes et al., 2022) and Total Coding Rate (TCR) (Yu et al., 2020; Tong et al., 2023) enforce orthogonality among negative pairs via soft regularization, while W-MSE (Ermolov et al., 2021) employs feature whitening. They align with

OLE's high-rank constraints, but neglect the low-rank embedding of positive pairs. This gap prompts the question: Why has OLE not been fully integrated into SSL?

Our investigation reveals two key challenges. First, in SSL, each instance acts as its own class, creating an infinite number of "classes." The OLE objective requires enforcing orthogonality between classes and minimizing intra-class ranks, but orthogonality is mathematically impossible with infinite classes in a limited feature space, leading to representational collapse. Second, enforcing low-rank constraints on feature vectors can fail to distinguish between positively and negatively correlated features. The nuclear norm ignores the signs of cosine similarities, which is critical to aligning positive pairs in SSL, where supervision is absent.

To address these issues, we propose **SSOLE** (Self-Supervised Orthogonal Low-rank Embedding), a framework that effectively integrates OLE into SSL. Our approach involves two key strategies: (1) **Decoupling low-rank and high-rank enforcement**: We separately manage positive pair attraction and negative pair repulsion, eliminating the dependence of low-rank enforcement on inter-class orthogonality. We optimize both the lower and upper bounds of the contrastive loss using nuclear norms to adapt OLE for SSL. (2) **Applying low-rank constraints to feature deviations**: Instead of directly applying low-rank constraints to the feature vectors, we apply them to the deviations from their mean, ensuring that the nuclear norm captures the signs of cosine similarities and aligns positive pairs correctly.

Through theoretical analysis and empirical evaluations, we demonstrate that these adaptations are essential for SSOLE's success in SSL. Our method addresses the identified challenges, achieving state-of-the-art performance on benchmark datasets. SSOLE leverages OLE's strengths while avoiding its limitations in SSL, offering an efficient solution for representation learning.

In summary, our contributions are: (1) identifying key challenges in applying OLE to SSL and offering insights into its limitations; (2) proposing SSOLE, which decouples low-rank and high-rank enforcement and applies low-rank constraints to feature deviations; (3) validating SSOLE's superior performance through theoretical and empirical analysis; and (4) opening new avenues for matrix-level operations and rank constraints in SSL, offering potential for future advancements.

## 2 BACKGROUND

### 2.1 ORTHOGONAL LOW-RANK EMBEDDING (OLE)

Orthogonal Low-rank Embedding (OLE) (Lezama et al., 2018) was introduced as a geometric loss function to improve deep network representations by simultaneously minimizing intra-class variance and maximizing inter-class separability. Unlike traditional softmax-based classification losses, OLE enforces that samples from the same class lie in a low-rank subspace while ensuring that different class subspaces are orthogonal. This approach encourages compact and well-separated representations in the embedding space, leading to improved generalization and robustness.

The OLE loss is defined as:

$$\mathcal{L}_{\text{OLE}} = \sum_{c=1}^{K} \|\Phi(\mathbf{X}_c)\|_* - \|\Phi(\mathbf{X})\|_*, \tag{1}$$

where $\mathbf{X}_c$ denotes the set of samples from class $c$, $\mathbf{X}$ is the entire data set from $K$ classes, $\Phi(\mathbf{X})$ represents the transformation applied to the entire dataset, and $\|\cdot\|_*$ is the nuclear norm, which approximates the rank of a matrix. The first term minimizes the intra-class rank, while the second term maximizes the inter-class separability by encouraging orthogonality between different classes.

OLE achieves a non-negative value and reaches zero when the representations of all classes are orthogonal, implying maximal inter-class separation. At the same time, intra-class samples are compressed into a low-dimensional subspace, preserving intra-class similarity. The nuclear norm serves as a convex relaxation of the rank function, facilitating efficient optimization through gradient-based methods.

## 3 LIMITATIONS OF OLE FOR SSL

Although OLE has been successful in supervised settings, its application to SSL poses significant challenges. These challenges stem directly from two inherent limitations of OLE, which are manageable in the presence of labels but become problematic in SSL due to the lack of supervision. Firstly, in OLE, low-rank (intra-class similarity) and high-rank (inter-class dissimilarity) constraints are deeply intertwined, making it difficult to optimize them independently. Secondly, the nuclear norm used in OLE cannot differentiate between positively and negatively correlated vectors. In the following subsections, we elaborate on the resulted challenges from them.

### 3.1 VIRTUALLY INFINITE NUMBER OF "CLASSES" IN SSL

A key challenge of applying OLE to self-supervised learning (SSL) arises from the fact that, in SSL, each instance is treated as its own class. This results in a virtually infinite number of "classes," making it impossible to achieve orthogonality among all class representations. The entanglement of low-rank and high-rank enforcement exacerbates this issue, as the original OLE objective depends on achieving inter-class orthogonality before minimizing intra-class ranks.

In supervised settings with finite classes, OLE works by first ensuring orthogonality between different class subspaces and then minimizing the intra-class rank, leading to compact and discriminative features. However, in SSL, where orthogonality is not feasible, OLE "cheats" by reducing vector length $l$ and the angle $\theta$ between vectors to minimize the nuclear norm, resulting in representational collapse.

To explain this, consider the model generating random representations from a $d$-dimensional Gaussian distribution. For $m$ samples drawn from this distribution, the expected nuclear norm of the matrix $V$ of these samples can be approximated by the following lemma:

**Lemma 3.1.** *Let $V$ be an $m \times d$ matrix, with $d \gg 1$, whose rows are sampled from a $d$-dimensional Gaussian distribution $\mathcal{N}\left(\frac{s\mathbf{1}_d}{\sqrt{d}}, \frac{s^2\sigma^2\mathbf{I}_d}{d}\right)$. The expected nuclear norm of $V$ satisfies:*

$$\mathbb{E}[\|V\|_*] = \begin{cases} s \cdot (\sigma \cdot \mathcal{O}(m) + \mathcal{O}(\sqrt{m})), & \text{if } m \ll d, \\ s \cdot \left(\sigma \cdot \mathcal{O}\left(\sqrt{md}\right) + \mathcal{O}(\sqrt{m})\right), & \text{if } m \gg d. \end{cases}$$

Lemma 3.1 follows the random matrix theory, and the proof is provided in Appendix A.1. Extending it to OLE loss over $B$ images, each with $N$ views, we can approximate the OLE loss.

**Theorem 3.2.** *If the model's output representation conforms to $\mathcal{N}\left(\frac{s\mathbf{1}_d}{\sqrt{d}}, \frac{s^2\sigma^2\mathbf{I}_d}{d}\right)$, then for $B$ images, each with $N$ views, where $1 < N \ll d \ll BN$, the OLE loss satisfies:*

$$\mathcal{L}_{OLE} = s\sqrt{BN}\left(\sigma \cdot \mathcal{O}\left(\sqrt{BN}\right) + \mathcal{O}\left(\sqrt{B}\right)\right).$$

The proof is provided in Appendix A.2. Theorem 3.2 shows that $\mathcal{L}_{OLE}$ is negatively correlated with the scale $s$ and $\sigma$. The analysis in Appendix A.3 also shows how $s$ and $\sigma$ relate the average vector length $l$ and angle $\theta$ between vectors. So the OLE loss decreases as $l$ or $\theta$ becomes smaller. In SSL, where orthogonality is unachievable, OLE reduces $l$ and $\theta$ to minimize the loss, leading to collapsed, trivial representations. Once this collapse occurs, further training becomes difficult, as the nuclear norm is dominated by the reduction in $l$ and $\theta$, preventing the model from learning separable features.

Even normalization techniques in SSL, which prevent $l$ from shrinking, do not solve the problem. The OLE loss can still minimize $\theta$, leading to collapse. Additionally, imposing a bound on the intra-class nuclear norm, as done in the original OLE, does not resolve this. If the bound is not reached, the model continues to shrink both $l$ and $\theta$. Once the bound is reached, the model oscillates between increasing and shrinking $l$ and $\theta$, causing instability.

This challenge highlights a fundamental limitation for OLE. While OLE performs well in supervised settings by enforcing orthogonality, it struggles in SSL, where orthogonality cannot be achieved. Despite efforts like normalization and bounded nuclear norms, there is currently no effective solution to this issue. To address this, Section 4.1 proposes to decouple low-rank and high-rank enforcements, allowing independent optimization of them to avoid representational collapse. It also explores the connection between nuclear norm optimization and contrastive objectives to refine these enforcements.

## 3.2 LOW-RANK REPRESENTATIONS MAY NOT BE GOOD REPRESENTATIONS

Another significant challenge arises from the nuclear norm's inability to distinguish between positively and negatively correlated features. In SSL, this limitation leads to misalignment between positive pairs because the nuclear norm treats aligned and anti-aligned vectors similarly, degrading the quality of learned representations. Without distinguishing between aligned and anti-aligned vectors, the nuclear norm can treat opposite directions similarly, causing misaligned positive pairs and undermining the learning process. This issue is particularly problematic in SSL, where the lack of labels prevents the model from correcting these misalignment.

Consider the nuclear norm for a 2-row matrix $V$ where the rows are two unit vectors separated by an angle $\theta$. Its nuclear norm is given by $\|V\|_* = \sqrt{1 + \cos\theta} + \sqrt{1 - \cos\theta}$. This shows that the nuclear norm only depends on the magnitude of $\cos\theta$, regardless of whether the cosine is positive or negative. Whether the vectors are aligned ($\cos\theta > 0$) or anti-aligned ($\cos\theta < 0$), the nuclear norm remains unchanged. This inability to distinguish between the directions of the vectors is problematic for SSL, where positive pairs should be aligned and negative pairs should remain distinct.

More generally, this limitation can be formalized. Let $V$ be an $m$-row matrix where each row is a $d$-dimensional unit vector, and let $P$ be an $m \times m$ diagonal matrix whose diagonal elements are either $1$ or $-1$, which invert or preserve the signs of the rows of $V$. The nuclear norm of $PV$ remains unchanged, demonstrating the unitarily invariant property of the nuclear norm. Further discussion is provided in Appendix A.4.

Currently, there are no effective methods to resolve this limitation in SSL. As a result, the inability of the nuclear norm to differentiate vector directions remains a significant obstacle for OLE-based methods in SSL. To address this challenge, Section 4.2 proposes using deviation matrices for low-rank enforcement. This ensures alignment of positive pairs by focusing on deviations from the mean and circumventing the nuclear norm's insensitivity to cosine similarity signs.

## 4 METHOD

We present how to address the key challenges of applying OLE to SSL. To handle the issue of infinite classes in SSL, we decouple low-rank and high-rank enforcement, allowing independent control of positive pair alignment and negative pair separation. We also modify the nuclear norm enforcement to account for the signs of cosine similarities, preventing representational collapse and ensuring proper alignment of feature spaces. Our framework, SSOLE (Self-Supervised Orthogonal Low-rank Embedding), integrates these solutions to effectively adapt OLE for SSL tasks.

### 4.1 ADDRESSING CHALLENGE 1: DECOUPLING LOW-RANK AND HIGH-RANK ENFORCEMENT

To address the challenge of infinite number of "classes" in SSL , we propose to decouple low-rank and high-rank enforcement to ensure that each term operates independently, allowing the model to enforce both alignment within positive pairs and uniformity across negative pairs more effectively.

**Normalizing feature vectors**: First, we alleviate some of the collapse problems by normalizing the feature vectors. By ensuring that all feature vectors have a fixed unit length, we prevent the model from shrinking vector lengths to minimize the loss artificially.

**Decoupling low-rank and high-rank enforcement**: To further address the challenge, we modify the original OLE objective to separate the low-rank enforcement for positive pairs from the high-rank enforcement for negative pairs. We propose the following loss function:

$$\mathcal{L} = \frac{1}{B}\sum_{b=1}^{B} h_1(\|\mathbf{Z}_{b,:}\|_*, N) + \lambda\frac{1}{N}\sum_{n=1}^{N} h_2(\|\mathbf{Z}_{:,n}\|_*, B), \tag{2}$$

where $\mathbf{Z} \in \mathbb{R}^{B \times N \times d}$ is the representation tensor for $B$ images in a batch, each with $N$ augmented views. The matrix $\mathbf{Z}_{b,:}$ represents the views of the $b^{th}$ image, and $\mathbf{Z}_{:,n}$ represents the $n^{th}$ view across all images. $h_1(\cdot)$ and $h_2(\cdot)$ are strictly increasing and decreasing functions respectively and used to control the nuclear norm-based enforcement for low-rank and high-rank. The term $\lambda$ balances the two objectives.

This formulation decouples the two terms and allows each objective to be enforced more appropriately, reducing the entanglement between intra-class and inter-class representations that causes collapse.

However, directly applying the nuclear norm as the function $h_1(\cdot)$ or $h_2(\cdot)$ still carries the risk of collapse. To address this, we explore relationships between nuclear norms and cosine similarities, drawing inspiration from prior work (Wang & Isola, 2020), which decomposes SSL objectives into alignment (for positive pairs) and uniformity (for negative pairs).

**Studying the nuclear norms**. To guide the choice of $h_1(\cdot)$ and $h_2(\cdot)$, we develop the following theorem, which relates nuclear norms to the cosine similarities between vectors.

**Theorem 4.1.** *For a set of $N$ $d$-dimensional unit vectors $\{\boldsymbol{v}_i\}$, where $d > N$, the nuclear norm of the matrix $\boldsymbol{V}$, containing the vectors as columns, is bounded by:*

$$\sqrt{\frac{N}{|\cos(\theta)|}} \leq \|\boldsymbol{V}\|_* \leq \sqrt{N} \cdot \sqrt{\overline{\cos(\theta)}} + \sqrt{N(N-1)} \cdot \sqrt{1 - \overline{\cos(\theta)}},$$

*where $\overline{\cos(\theta)} = \frac{1}{N^2}\sum_{i=1}^{N}\sum_{j=1}^{N}\cos(\theta_{i,j}) = \frac{1}{N^2}\sum_{i=1}^{N}\sum_{j=1}^{N}\boldsymbol{v}_i^T\boldsymbol{v}_j$ is the average cosine similarity between every two vectors, and $\overline{|\cos(\theta)|}$ is the average absolute cosine similarity.*

*The lower bound is attained when the vectors can be equally grouped into $\frac{1}{|\cos(\theta)|}$ groups such that within each group, the vectors are either perfectly aligned or anti-aligned, while the vectors from different groups are orthogonal. The upper bound is satisfied when the angles are identical for all vector pairs $(\boldsymbol{v}_i, \boldsymbol{v}_j)$ where $i \neq j$.*

The proof is provided in Appendix A.5. Theorem 4.1 shows that the nuclear norm is both lower and upper bounded by a function of the average (absolute) cosine similarity between vectors. When $\overline{\cos(\theta)} = \overline{|\cos(\theta)|} = \frac{1}{N}$, both the lower and the upper bounds reach the maximum of $N$. In contrast, when $\overline{\cos(\theta)} = \overline{|\cos(\theta)|} = 1$, both the lower and upper bounds attain the minimum of $\sqrt{N}$. These bounds allow us to design functions $h_1(\cdot)$ and $h_2(\cdot)$.

Inverting the bounds in Theorem 4.1, we can derive lower and upper bounds for the average absolute cosine similarity $\overline{|\cos(\theta)|}$ in terms of the nuclear norm:

$$\frac{N}{\|\boldsymbol{V}\|_*^2} \leq \overline{|\cos(\theta)|}; \quad \overline{\cos(\theta)} \leq \cos^2\left(\arcsin(\frac{\|\boldsymbol{V}\|_*}{N}) - \arcsin(\frac{1}{\sqrt{N}})\right). \tag{3}$$

This insight enables us to control the cosine similarities between vectors through nuclear norm minimization or maximization.

**Caution in Low-rank Enforcement**: However, care must be taken when using these bounds for low-rank enforcement. The lower bound on $\overline{|\cos(\theta)|}$ ignores the signs of the cosine similarities, and the upper bound has a local minimum at $\overline{\cos(\theta)} = 0$ where all off-diagonal cosine similarities are $\frac{-1}{N-1}$ which is negative. This is not an issue for high-rank enforcement, as the goal is to achieve orthogonality. However, for low-rank enforcement, there may be cases where the absolute cosine similarity is maximized, but the actual cosine similarity averages to zero, resulting in misaligned vectors. Thus, we need to be cautious in designing $h_1(\cdot)$ to ensure it aligns positive pairs effectively.

## 4.2 ADDRESSING CHALLENGE 2: LOW-RANK ENFORCEMENT VIA DEVIATION MATRICES

To address the challenge of misalignment between positive pairs , we propose enforcing low-rank constraints on a **deviation matrix**. The core issue lies in the nuclear norm's insensitivity to cosine similarity signs. Applying the nuclear norm directly to positive pairs' representations risks misaligning them. By subtracting the mean vector from each individual vector, we create a deviation matrix that captures how each vector deviates from the average. The rank of the deviation matrix approximates the rank of the original matrix (with at most a difference of 1), so low-rank enforcement on the deviation matrix remains valid. Additionally, deviations in opposite directions are acceptable in SSL, meaning this approach circumvents the cosine similarity sign issue.

We formalize this approach with the following theorem:

**Theorem 4.2.** *For a set of $N$ $d$-dimensional unit vectors $\{v_i\}$, where $d > N$, the nuclear norm of $\tilde{V} = V - \bar{v}\mathbf{1}^\top$, where $\bar{v} := \frac{1}{N}\sum_{i=1}^{N} v_i$ and $\mathbf{1} \in \mathbb{R}^N$ is a vector of ones, is bounded by:*

$$\sqrt{N} \cdot \sqrt{1 - \overline{\cos(\theta)}} \leq \|\tilde{V}\|_* \leq \sqrt{N(N-1)} \cdot \sqrt{1 - \overline{\cos(\theta)}}.$$

*The lower bound is reached when the vectors $\{v_i\}$ can be grouped into two perfectly aligned sets, and the upper bound holds when the angles are identical for all vector pairs $(v_i, v_j)$ where $i \neq j$.*

The proof is provided in Appendix A.6. The bounds provided by Theorem 4.2 depend on $\overline{\cos(\theta)}$, which incorporates the signs of the cosine similarities. This ensures that low-rank enforcement will not confuse with anti-aligned vectors, unlike in the original matrix where such signs may be ignored.

Further, when $\overline{\cos(\theta)}$ reaches its maximum value of 1 (perfect alignment), both the lower and upper bounds become 0, indicating a perfectly low-rank representation. Conversely, when $\overline{\cos(\theta)}$ reaches its minimum value of 0 (orthogonal or anti-aligned vectors), the lower bound becomes $\sqrt{N}$, corresponding to the lowest nuclear norm of the original matrix, while the upper bound reaches $\sqrt{N(N-1)}$, the highest possible nuclear norm for the deviation matrix.

By inverting the bounds in Theorem 4.2, we derive lower and upper bounds for $1 - \overline{\cos(\theta)}$ in terms of the nuclear norm of the deviation matrix:

$$\frac{\|\tilde{V}\|_*^2}{N(N-1)} \leq 1 - \overline{\cos(\theta)} \leq \frac{\|\tilde{V}\|_*^2}{N}. \tag{4}$$

These bounds offer insights into the cosine similarities based on the nuclear norm of the deviation matrix. Enforcing low-rank representations on the deviation matrix resolves the issues caused by applying the nuclear norm directly to the feature matrix, ensuring that low-rank enforcement does not collapse into suboptimal solutions. This approach allows SSL to maintain the benefits of low-rank enforcement while preserving meaningful feature representations.

### 4.3 TRAINING OBJECTIVE

Given the improvements to low-rank enforcement, we need to modify Equation (2) to derive the training objective for SSOLE as follows:

$$\mathcal{L}_{SSOLE} = \frac{1}{B}\sum_{b=1}^{B} h_1(\|\tilde{\mathbf{Z}}_{b,:}\|_*, N) + \lambda \frac{1}{N}\sum_{n=1}^{N} h_2(\|\mathbf{Z}_{:,n}\|_*, B), \tag{5}$$

where $\tilde{\mathbf{Z}}_{b,:}$ denotes the deviation matrix of $\mathbf{Z}_{b,:}$.
The loss $\mathcal{L}_{SSOLE}$ leverages both lower and upper bounds for intra-class and inter-class cosine similarity derived from Theorem 4.1 and Theorem 4.2. These bounds provide a framework for optimizing a contrastive-like objective based on nuclear norms, which aligns with the goals of maximizing intra-class similarity (alignment) and inter-class dissimilarity (uniformity).

For low-rank enforcement, we derive $h_1(\cdot)$ from Equation (4), basing on optimizing the lower bound of the nuclear norm. Since the lower and upper bounds are equivalent up to a factor of $\frac{1}{N-1}$, optimizing the lower bound is sufficient to optimize both bounds. Then $h_1(\cdot)$ is given by[1]:

$$h_1(\|\tilde{\mathbf{Z}}_{b,:}\|_*, N) = \frac{\|\tilde{\mathbf{Z}}_{b,:}\|_*^2}{(N-1)^2}. \tag{6}$$

For high-rank enforcement, we derive $h_2(\cdot)$ from Equation (3), optimizing the average of the lower and upper bounds of average (absolute) cosine similarity from the nuclear norm. $h_2(\cdot)$ is given by:

$$h_2(\|\mathbf{Z}_{:,n}\|_*, B) = \frac{B}{2(B-1)}\left(\frac{B}{\|\mathbf{Z}_{:,n}\|_*^2} + \cos^2(\arcsin(\frac{\|\mathbf{Z}_{:,n}\|_*}{B}) - \arcsin(\frac{1}{\sqrt{B}}))\right) - \frac{1}{B-1}. \tag{7}$$

The SSOLE training objective combines the advantages of OLE for SSL while optimizing both a lower and upper bound for a contrastive-like loss using nuclear norms. By utilizing these bounds, the model can enforce alignment and uniformity in a more efficient manner, avoiding the representational collapse seen in vanilla OLE approaches.

---

[1]Note that in $h_1(\cdot)$ and $h_2(\cdot)$, we need to multiply a factor of $\frac{N-1}{N}$ or $\frac{B-1}{B}$ to obtain the estimated average (absolute) cosine similarities for all vector pairs $(v_i, v_j)$ where $i \neq j$.

## 5 RELATED WORK

### 5.1 SELF-SUPERVISED LEARNING

Self-supervised learning has advanced significantly in the realm of image recognition by leveraging various innovative techniques. Methods such as BYOL (Grill et al., 2020) and MoCo (He et al., 2020) use bootstrapping and dynamic dictionaries to enhance representation learning. SimSiam (Chen & He, 2021) explores learning representations without negative pairs, while CPCv2 (Henaff, 2020) emphasizes data-efficient image recognition. The principles of alignment and uniformity on a hypersphere have been analyzed by Wang & Isola (2020). Meanwhile, the impact of view (augmentation) selection in contrastive learning is investigated by Tian et al. (2020). DINO (Caron et al., 2021) stands out as a robust self-supervised method, having evolved its training protocol to achieve competitive results. MoCo v3 (Chen et al., 2021) builds upon momentum contrast for training Vision Transformers. VicReg (Bardes et al., 2022) introduces an approach based on variance, invariance, and covariance, while Wang et al. Wang et al. (2021) address inefficiencies in representation learning. iBOT (Zhou et al., 2022) focuses on Image BERT pre-training and RELIC v2 (Tomasev et al., 2022) ambitiously aims to outperform supervised learning on ImageNet without labels. Furthermore, the work of He et al. (2021) introduces Masked Autoencoders (MAE), a scalable vision learner that benefits from the reconstruction of masked image patches. This method implicitly utilizes multiple views by treating visible and masked patches differently during the learning process. Wu et al. (2018) propose a non-parametric approach to instance discrimination.

Multi-View Self-Supervised Learning (MV-SSL) has recently emerged as a potent paradigm to harness the information from various augmentations or views of the same data. This approach has led to significant advancements in SSL by promoting more generalized feature representations. SwAV (Caron et al., 2020) introduces a unique "swapped prediction" task to SSL, utilizing cluster assignments as pseudo-labels to encourage consistency across different augmentations or views. It employs multiple views of an image to compute these assignments, promoting invariance across views and improving the learned representations. LORAC (Wang et al., 2022) extends the principles of MoCo by incorporating low-rank embedding as a prior, which is particularly beneficial for SSL. It leverages multiple views to enforce consistency. EMP-SSL (Tong et al., 2023) takes a different approach by generating an extremely large number of patches or views from the input images, significantly reducing the training eopchs to converge.

This work is related to MV-SSL and LORAC. While LORAC incorporates low-rank constraints as a regularization term within contrastive learning frameworks, it does not fully exploit the potential of OLE as a central metric. In contrast, our proposed SSOLE redefines the role of OLE in SSL by using it as an intrinsic metric for both positive alignment and negative separation. By strategically adapting OLE to address SSL-specific challenges, SSOLE fully leverages the unique properties of multi-view data. This represents a significant departure from prior approaches and unlocks the full potential of OLE in SSL.

### 5.2 ORTHOGONAL LOW-RANK EMBEDDING AND RELATED TECHNIQUES

The concept of learning low-dimensional, structured representations through low-rank constraints has been extensively studied across multiple domains, including Principal Component Analysis (PCA), Linear Discriminant Analysis (LDA) (Hastie et al., 2009), face recognition (Yang et al., 2016; Xue et al., 2017; Lezama et al., 2017; Xue et al., 2019), and image classification(Zhang et al., 2013; Jiang et al., 2014; Zhang et al., 2016). High-rank regularization has also been explored for applications such as learning orthogonal projections in deep networks (Vorontsov et al., 2017), improving recurrent network performance (Bansal et al., 2018), and capsule subspace projection (Zhang et al., 2018).

In deep learning, Orthogonal Low-Rank Embedding (OLE) (Lezama et al., 2018) extended these ideas by introducing low-rank and orthogonal constraints into supervised classification tasks. By enforcing intra-class low-rank and inter-class orthogonality, OLE achieves compact and discriminative representations. Inspired by these principles, subsequent works like LORAC(Wang et al., 2022) incorporated low-rank priors into SSL, albeit as regularizations rather than direct metrics.

Further extensions of OLE include data-dependent regularizations (Zhu et al., 2019), which aim to enhance pattern discovery and prevent overfitting, and Meta-OLE (Wang et al., 2023), which adapts OLE for meta-learning.

While these works establish the utility of low-rank and high-rank constraints, they primarily focus on supervised settings. Their direct application to SSL faces unique challenges. Our proposed method, SSOLE, builds upon OLE by addressing these challenges through two key innovations. These adaptations enable SSOLE to effectively extend OLE principles to SSL.

# 6 EXPERIMENTS

## 6.1 ABLATION STUDIES

We focus on exploring effective adaptations of OLE for SSL. The ResNet-18 (He et al., 2016) architecture is employed on the ImageNet100 (Deng et al., 2009) dataset. Detailed information on data augmentation and training procedures is provided in Appendix C.

### 6.1.1 ADAPTING OLE TO SSL

We introduce a baseline method of InfoNCE-M, an extension of the InfoNCE loss adapted for MV-SSL. InfoNCE-M uses the mean of all views as the anchor for each image, computing the InfoNCE loss for each view and averaging these values. The formula for InfoNCE-M is

$$\mathcal{L}_{\text{InfoNCE-M}} = -\frac{1}{BN}\sum_{i=1}^{B}\sum_{j=1}^{N}\log\frac{e^{\text{sim}(\mathcal{Z}_{i,j},\mathbf{m}_i)/\tau}}{\sum_{k=1}^{B}e^{\text{sim}(\mathcal{Z}_{i,j},\mathbf{m}_k)/\tau}}, \tag{8}$$

where $\mathbf{m}_i = \frac{1}{N}\sum_{j=1}^{N}\mathcal{Z}_{i,j}$ is the mean embedding (anchor) of all views for the $i$-th image, and $\tau$ is the temperature scaling parameter. The function $\text{sim}(\cdot,\cdot)$ computes the cosine similarity.

We then explore adaptations of OLE for SSL. The results of these adaptations are summarized in Table 1. The standard $\mathcal{L}_{\text{InfoNCE-M}}$ with a hyperparameter ($\tau = 0.2$) sets a baseline with a respectable Top-1 accuracy of 76.4% and Top-5 accuracy of 93.0%. Then we observe that the direct application of $\mathcal{L}_{\text{OLE}}$ faced convergence issues. When we normalize the representation vectors, the training collapses. This indicates inherent chal-

Table 1: Studying adaptations of OLE for SSL using various strategies. We use 5 views per image. all models are trained for 100 epochs.

| Objective | h.param. | Top-1 | Top-5 |
|---|---|---|---|
| $\mathcal{L}_{\text{InfoNCE-M}}$ | $\tau = 0.2$ | 76.4 | 93.0 |
| $\mathcal{L}_{\text{OLE}}$ | - | failed to converge | |
| + normalization | - | collapse | |
| + loss decoupling | $\lambda = 2.15$ | 43.2 | 71.2 |
| + enhanced low-rank ($\mathcal{L}_{\text{SSOLE}}$) | $\lambda = 0.7$ | **78.5** | **94.4** |

lenges in applying OLE to SSL without suitable modifications. Decoupling the low-rank and high-rank enforcement, but without using the deviation matrix for intra-class alignment, helps to stabilize the training but has bad performance, and it showed high sensitivity to the hyperparameter $\lambda$. Specifically, training tended to collapse to constants for $\lambda \le 2.10$ and to random values for $\lambda \ge 2.20$. A temporary stabilization occurred at $\lambda = 2.15$, but the training loss eventually diverged after about 30 epochs, leading to SVD errors. This instability indicates the difficulties in directly applying low-rank constraints to the original embedding matrix.

Table 2: Impact of $\lambda$ on linear probing top-1 Accuracy (%).

| $\lambda$ | 0.1 | 0.2 | 0.3 | 0.4 | 0.5 | 0.6 | 0.7 | 0.8 | 0.9 | 1.0 | 1.1 | 1.2 | 1.3 | 1.4 | 1.5 | 1.6 | 1.7 | 1.8 | 1.9 | 2.0 | 2.5 | 3.0 |
|---|---|---|---|---|---|---|---|---|---|---|---|---|---|---|---|---|---|---|---|---|---|---|
| Acc. | 72.1 | 73.9 | 76.5 | 77.5 | 78.4 | 78.5 | 78.5 | 78.4 | 78.5 | 78.0 | 78.3 | 78.6 | 78.3 | 78.3 | 78.1 | 78.3 | 78.4 | 78.3 | 78.4 | 78.0 | 77.3 | 77.3 |

In contrast, the $\mathcal{L}_{\text{SSOLE}}$ loss with further enhanced low-rank enforcement demonstrated remarkable insensitivity to the value of $\lambda$, which was varied between 0.5 and 1.9, shown in Table 2. For smaller values of $\lambda \in [0.1, 0.4]$, the model prioritizes intra-class compactness, which can lead to under-penalized inter-class overlap, slightly degrading performance. For larger values of $\lambda \in [2.0, 3.0]$, the model emphasizes inter-class separability, sometimes at the expense of intra-class consistency, resulting in over-dispersed features. Optimal performance was achieved around $\lambda = 0.7$, with top-1 accuracy of 78.5% and top-5 accuracy of 94.4%. This relative robustness of $\mathcal{L}_{\text{SSOLE}}$ highlights its suitability for SSL tasks and verifies the effectiveness of our approach in adapting OLE to the SSL framework.

### 6.1.2 STUDIES ON THE NUMBER OF VIEWS

Understanding the impact of the number of views on model performance is crucial in SSL. As shown in Figure 1, both models exhibit an increase in Top-1 accuracy with more views, but SSOLE consistently outperforms InfoNCE-M. SSOLE's performance notably improves up to 8 views, after which it plateaus, suggesting that the intrinsic rank of views from the same instance is likely less than 8. This plateau indicates diminishing returns beyond that point. SSOLE's consistently superior performance highlights its effectiveness in utilizing additional views, while InfoNCE-M shows more modest gains, underscoring its relative inefficiency in leveraging extra views.

Figure 1: Comparison between InfoNCE-M and SSOLE with various numbers of views.

## 6.2 COMPARISON ON IMAGENET100

Table 3 provides a detailed comparison of various methods on the ImageNet100 dataset, using ResNet-18 as the backbone. Our proposed method, SSOLE, achieves the highest Top-1 accuracy of 82.5%, outperforming established methods such as MoCo-M, SwAV, VICReg, BYOL, LORAC, and EMP-SSL. This result highlights SSOLE's superior performance, especially considering its modest computational demands.

In contrast to methods like BYOL and VICReg, which require large batch sizes, SSOLE achieves higher performance with a batch size of only 128. Additionally, SSOLE employs a single-branch encoder, while methods like MoCo, BYOL, and LO-RAC use an EMA teacher model. Lastly, unlike MoCo and LORAC, which rely on large memory banks, SSOLE avoids additional memory requirements. We utilize 8 crops per image, consisting of 4 large and 4 small crops.

Table 3: Comparative performance on ImageNet100. The table includes batch size (bs), number of epochs, number of crops, number of forward passes, and Top-1 accuracy (%).

| Method | bs | #epochs | #crops | #forwards | Top-1 |
|---|---|---|---|---|---|
| MoCo-M (Wang et al., 2022) | 128 | 100 | 8 | 10 | 77.0 |
| SwAV (Caron et al., 2020) | 256 | 400 | 8 | 8 | 74.0 |
| VICReg (Bardes et al., 2022) | 2048 | 400 | 2 | 2 | 79.2 |
| BYOL (Grill et al., 2020) | 4096 | 400 | 2 | 4 | 80.2 |
| LORAC (Wang et al., 2022) | 128 | 100 | 8 | 10 | 78.7 |
| EMP-SSL (Tong et al., 2023) | 100 | 10 | 200 | 200 | 78.9 |
| INTL (Weng et al., 2024) | 128 | 400 | 2 | 2 | 81.7 |
| SSOLE (Ours) | 128 | 100 | 8 | 8 | **82.5** |

Details on our multi-crop strategy are provided in Appendix C.1. For a fair comparison, we train SSOLE for just 100 epochs, ensuring that the number of forward passes per iteration (#forwards) × the number of epochs (#epochs) is equal to or less than those of other methods.

## 6.3 EVALUATION ON FULL IMAGENET-1K

In this subsection, we assess the performance of our proposed SSOLE method when evaluated on the full ImageNet-1k dataset. We follow a similar evaluation protocol as used for ImageNet100, with the addition of semi-supervised learning settings where only a fraction of the labels are available.

### 6.3.1 SELF-SUPERVISED AND SEMI-SUPERVISED PERFORMANCE

Table 4 showcases the performance of various methods on the full ImageNet dataset, with a special emphasis on the SSOLE method here proposed. The table reports both linear probing and semi-supervised learning accuracies, highlighting the efficacy of SSOLE in different learning regimes.

SSOLE demonstrates superior overall performance, particularly when compared to LORAC, which also leverages multiple views and employs low-rank embedding as a regularization prior. While LORAC achieves a respectable Top-1 accuracy of 73.2% and Top-5 accuracy of 91.6%, SSOLE surpasses it with a Top-1 accuracy of 73.9% and a Top-5 accuracy of 91.7%. Although SSOLE slightly underperforms INTL, which uses an EMA teacher, it is important to note that SSOLE does not rely on dual-branch encoders or EMA. Moreover, SSOLE significantly outperforms INTL when EMA is not applied. In semi-supervised learning, the advantage of SSOLE becomes even more pronounced. SSOLE outperforms LORAC by over 5 percentage points in the 1% labeled data setting and approximately 2 percentage points in the 10% labeled data setting. These improvements underscore the effectiveness of integrating OLE directly as a metric, rather than using it merely as a regularization term, as is the case with LORAC.

Table 4: Performance on full ImageNet of different methods. The table reports Top-1 and Top-5 accuracies (%) for linear probing and semi-supervised settings with 1% and 10% labeled data.

| Method | BS | #Epochs | #Crops | #Forwards | Linear Probing Top-1 | Linear Probing Top-5 | Semi-supervised (1%) Top-1 | Semi-supervised (1%) Top-5 | Semi-supervised (10%) Top-1 | Semi-supervised (10%) Top-5 |
|---|---|---|---|---|---|---|---|---|---|---|
| Supervised | - | - | - | - | 76.5 | - | 25.4 | 56.4 | 48.4 | 80.4 |
| MoCo v2 (Chen et al., 2020b) | 256 | 200 | 2 | 2 | 71.1 | - | - | - | - | - |
| SimCLR (Chen et al., 2020a) | 4096 | 1000 | 2 | 2 | 69.3 | 89.0 | 48.3 | 75.5 | 65.6 | 87.8 |
| SimSiam (Chen & He, 2021) | 256 | 800 | 2 | 2 | 71.3 | - | - | - | - | - |
| Barlow Twins (Zbontar et al., 2021) | 2048 | 1000 | 2 | 2 | 73.2 | 91.0 | 55.0 | 79.2 | 69.7 | 89.3 |
| VICReg (Bardes et al., 2022) | 2048 | 1000 | 2 | 2 | 73.2 | 91.1 | 54.8 | 79.4 | 69.5 | 89.5 |
| BYOL (Grill et al., 2020) | 4096 | 400 | 2 | 4 | 73.2 | - | - | - | - | - |
| SwAV (Caron et al., 2020) | 256 | 200 | 8 | 10 | 72.7 | 91.5 | 49.6 | 76.1 | 67.7 | 88.7 |
| LORAC (Wang et al., 2022) | 256 | 200 | 8 | 10 | 73.2 | 91.6 | 50.0 | 76.3 | 68.0 | 88.9 |
| MEC (Liu et al., 2022) | 256 | 400 | 2 | 4 | 73.5 | - | - | - | - | - |
| Matrixl-SSL (Zhang et al., 2024) | 256 | 400 | 2 | 4 | 73.6 | - | - | - | - | - |
| INTL (Weng et al., 2024) | 512 | 800 | 2 | 2 | 73.1 | - | 55.0 | **80.8** | 69.4 | 89.8 |
| INTL (EMA) (Weng et al., 2024) | 256 | 800 | 2 | 2 | **74.3** | - | - | - | - | - |
| SSOLE (Ours) | 256 | 200 | 8 | 8 | 73.9 | **91.7** | **55.4** | 79.6 | **70.3** | **90.3** |

### 6.3.2 TRANSFERRING TO OTHER DATASETS

In this section, we evaluate the adaptability and robustness of the SSOLE framework through transfer learning. We apply the feature extractor trained on ImageNet and fine-tune on MS-COCO (Lin et al., 2015) for object detection and instance segmentation tasks. We also evaluate transfer learning to linear classification tasks on datasets including CIFAR10 (Krizhevsky & Hinton, 2009), CIFAR100 (Krizhevsky & Hinton, 2009), Aircraft (Maji et al., 2013), DTD (Cimpoi et al., 2014), and Flowers (Nilsback & Zisserman, 2008), each offering unique image content and complexity challenges. This diverse set enables a comprehensive assessment of how SSOLE's learned representations generalize across visual domains.

Table 5: Transfer Learning on object detection and instance segmentation on MS-COCO.

| Model | Object Detection | | | Instance Segmentation | | |
|---|---|---|---|---|---|---|
| | $AP_{50}$ | AP | $AP_{75}$ | $AP_{50}^{mk}$ | $AP^{mk}$ | $AP_{75}^{mk}$ |
| SimCLR | 57.7 | 37.9 | 40.9 | 54.6 | 33.3 | 35.3 |
| MoCo v2 | 58.9 | 39.3 | 42.5 | 55.8 | 34.4 | 36.5 |
| BYOL | 57.8 | 37.9 | 40.9 | 54.3 | 33.2 | 35.0 |
| SwAV | 58.6 | 38.4 | 41.3 | 55.2 | 33.8 | 35.9 |
| SimSiam | 59.3 | 39.2 | 42.1 | 56.0 | 34.4 | 36.7 |
| Barlow Twins | 59.0 | 39.2 | 42.5 | 56.0 | 34.3 | 36.5 |
| VICReg | - | 40.0 | - | - | - | 36.7 |
| MEC | 59.8 | 39.8 | 43.2 | 56.3 | 34.7 | 36.8 |
| Matrix-SSL | 60.8 | 41.0 | 44.2 | 57.5 | 35.6 | 38.0 |
| INTL | 61.0 | 41.0 | 44.5 | 57.7 | 35.6 | 37.8 |
| SSOLE (Ours) | **61.5** | **41.3** | **44.8** | **58.0** | **35.9** | **38.4** |

Table 5 shows that SSOLE outperforms all other methods on COCO object detection and instance segmentation, highlighting its robustness and adaptability; Table 6 illustrates SSOLE's superior performance in transfer settings compared to state-of-the-art methods like MoCov2, SwAV, and LORAC. SSOLE achieves the highest accuracy on CIFAR10, Aircraft, DTD, and Flowers, indicating its capabil-

Table 6: Transfer Learning on linear classification on various datasets.

| Method | CIFAR10 | CIFAR100 | Aircraft | DTD | Flowers |
|---|---|---|---|---|---|
| Supervised | 90.0 | 73.4 | 42.6 | 68.8 | 89.7 |
| MoCov2 | 89.8 | 71.0 | 39.3 | 69.2 | 87.4 |
| SwAV | 90.8 | 73.4 | 45.5 | 72.2 | 88.9 |
| LORAC | 91.8 | **75.3** | 47.8 | 72.7 | 89.5 |
| SSOLE (Ours) | **92.2** | 74.4 | **48.0** | **73.3** | **90.0** |

ity to capture generalizable and robust features. SSOLE's performance on CIFAR 10/100 highlights its ability to handle complex small-image classifications. Its success in the fine-grained classification on the Aircraft dataset and in diverse recognition tasks on DTD and Flowers further showcases its adaptability. Overall, SSOLE's consistent effectiveness across these varied datasets attests to its versatility and efficiency as a feature extractor.

## 7 CONCLUSION

In this paper, we presented SSOLE, which integrates OLE into the SSL paradigm. Our method addresses two critical challenges in applying OLE to SSL: the difficulty of enforcing orthogonality in the presence of an infinite number of classes, and the nuclear norm's inability to distinguish between positive and negative correlations. By decoupling low-rank and high-rank enforcement and applying low-rank constraints to feature deviations, SSOLE effectively adapts OLE for self-supervised tasks. Through extensive experiments, we demonstrated that SSOLE achieves state-of-the-art performance across linear probing, semi-supervised, and transfer learning tasks, all while maintaining computational efficiency. Notably, SSOLE achieves these results without relying on large batch sizes, memory banks, or complex architectures. SSOLE sets a new benchmark for integrating orthogonal low-rank representations into SSL, opening up promising directions for future research in SSL.

ACKNOWLEDGMENTS

Work partially supported by ONR, NSF, Simons Foundation, and gifts/awards from Google, Amazon, and Apple.

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

# A   PROOFS AND ANALYSIS

## A.1   PROOF FOR LEMMA 3.1

*Proof.* We begin by centering the matrix $\boldsymbol{V}$ to create a zero-mean matrix $\tilde{\boldsymbol{V}}$ as follows:

$$\tilde{\boldsymbol{V}} = \boldsymbol{V} - \frac{s\boldsymbol{1}_d^T}{\sqrt{d}}.$$

This gives rows of $\tilde{\boldsymbol{V}}$ sampled from $\mathcal{N}\left(0, \frac{s^2\sigma^2\boldsymbol{I}_d}{d}\right)$. We first compute the nuclear norm for $\tilde{\boldsymbol{V}}$ and then correct for the non-centered matrix.

**Cosine similarity of vectors.**

Let $\tilde{\boldsymbol{v}}_i$ and $\tilde{\boldsymbol{v}}_j$ be two distinct rows of $\tilde{\boldsymbol{V}}$, sampled from $\mathcal{N}(0, \frac{s^2\sigma^2\boldsymbol{I}_d}{d})$. The cosine similarity between these vectors is given by:

$$\cos(\theta_{i,j}) = \frac{\langle \tilde{\boldsymbol{v}}_i, \tilde{\boldsymbol{v}}_j \rangle}{\|\tilde{\boldsymbol{v}}_i\|\|\tilde{\boldsymbol{v}}_j\|} = \langle \frac{\tilde{\boldsymbol{v}}_i}{\|\tilde{\boldsymbol{v}}_i\|}, \frac{\tilde{\boldsymbol{v}}_j}{\|\tilde{\boldsymbol{v}}_j\|} \rangle.$$

Then both $\frac{\tilde{\boldsymbol{v}}_i}{\|\tilde{\boldsymbol{v}}_i\|}$ and $\frac{\tilde{\boldsymbol{v}}_j}{\|\tilde{\boldsymbol{v}}_j\|}$ follow a uniform probability distribution over the unit sphere $\mathcal{S}^{d-1}$ (Muller, 1959; Marsaglia, 1972).

Mardia & Jupp (2009) shows that $\frac{1+\cos(\theta_{i,j})}{2}$ follows a Beta distribution, $\text{Beta}(\frac{d-1}{2}, \frac{d+1}{2})$, and

$$\mathbb{E}[\cos(\theta_{i,j})] = 0,$$
$$\text{Var}[\cos(\theta_{i,j})] = \frac{1}{d}.$$

Using Chebyshev's inequality, we bound the tail probability of $|\cos(\theta_{i,j})|$ for a given threshold $\frac{1}{d}$:

$$P\left(|\cos(\theta_{i,j})| > \frac{C}{\sqrt{d}}\right) \le \frac{1}{C^2}, \tag{9}$$

where $C$ is a constant and satisfies $1 \ll C \ll \sqrt{d}$.

**Expected nuclear norm for $m \ll d$.**

The nuclear norm of $\tilde{\boldsymbol{V}}$ depends on the norm of each row and cosine similarity between them.

When the rows of $\tilde{\boldsymbol{V}}$ are perfectly orthogonal, $\tilde{\boldsymbol{V}}$ achieves the maximum nuclear norm for given row norms, then $\tilde{\boldsymbol{V}}\tilde{\boldsymbol{V}}^T$ is an $m \times m$ diagonal matrix where the $i^{th}$ diagonal element is $\tilde{\boldsymbol{v}}_i\tilde{\boldsymbol{v}}_i^T = \|\tilde{\boldsymbol{v}}_i\|^2$. Then for $i = 1, \ldots, m$, $\tilde{\boldsymbol{V}}\tilde{\boldsymbol{V}}^T$ has an eigenvalue of $\|\tilde{\boldsymbol{v}}_i\|^2$; $\tilde{\boldsymbol{V}}$ has a corresponding singular value of $\|\tilde{\boldsymbol{v}}_i\|$. This determines the upper bound for the expected nuclear norm:

$$\mathbb{E}[\|\tilde{\boldsymbol{V}}\|_*] \le m \cdot \mathbb{E}[\|\tilde{\boldsymbol{v}}\|].$$

Since $\tilde{\boldsymbol{v}}$ follows $\mathcal{N}\left(0, \frac{s^2\sigma^2\boldsymbol{I}_d}{d}\right)$, $\|\frac{\sqrt{d}}{s\sigma}\tilde{\boldsymbol{v}}\|$ follows the chi distribution with $d$ degrees of freedom, *i.e.* $\mathcal{X}_d$. Therefore,

$$\mathbb{E}[\|\frac{\sqrt{d}}{s\sigma}\tilde{\boldsymbol{v}}\|] = \sqrt{2}\frac{\Gamma(\frac{d+1}{2})}{\Gamma(\frac{d}{2})},$$
$$\text{Var}[\|\frac{\sqrt{d}}{s\sigma}\tilde{\boldsymbol{v}}\|] = d - \mathbb{E}^2[\|\tilde{\boldsymbol{v}}\|].$$

Using the recurrence relation for the Gamma function and Stirling's approximation for large $d$, we can bound $\mathbb{E}[\|\frac{\sqrt{d}}{s\sigma}\tilde{\boldsymbol{v}}\|]$ as:

$$\mathbb{E}[\|\frac{\sqrt{d}}{s\sigma}\tilde{\boldsymbol{v}}\|] = \sqrt{d}\left(1 - \frac{1}{4d} + \mathcal{O}\left(\frac{1}{d^2}\right)\right) = \sqrt{d} + \mathcal{O}\left(\frac{1}{\sqrt{d}}\right).$$

Further,

$$\text{Var}[\|\frac{\sqrt{d}}{s\sigma}\tilde{\boldsymbol{v}}\|] = d - \left(d - \frac{1}{2} + \mathcal{O}\left(\frac{1}{d}\right)\right) = \frac{1}{2} + \mathcal{O}\left(\frac{1}{d}\right).$$

Thus,

$$\mathbb{E}[\|\tilde{\boldsymbol{v}}\|] = s\sigma + \mathcal{O}\left(\frac{1}{d}\right), \tag{10}$$

$$\text{Var}[\|\tilde{\boldsymbol{v}}\|] = \frac{s^2\sigma^2}{2d} + \mathcal{O}\left(\frac{1}{d^2}\right). \tag{11}$$

Therefore,

$$\mathbb{E}[\|\tilde{\boldsymbol{V}}\|_*] \le s \cdot \sigma \cdot m + \mathcal{O}\left(\frac{1}{d}\right). \tag{12}$$

Now, we derive the lower bound. $\|\tilde{\boldsymbol{V}}\|_*$ negatively relates to the absolute cosine similarity between its rows. Applying Equation (9), we derive the below probability:

$$P\left(\max_{i,j,i\neq j}|\cos(\theta_{i,j})| <= \frac{C}{\sqrt{d}}\right) \ge \left(1 - \frac{1}{C^2}\right)^{\frac{m(m-1)}{2}} = 1 - \mathcal{O}\left(\frac{1}{C^2}\right). \tag{13}$$

On the other hand, $\|\tilde{\boldsymbol{V}}\|_*$ positively relates to the norm of its rows.

Using Chebyshev's inequality with Equation (10) and Equation (11), we have:

$$P\left(|\|\tilde{\boldsymbol{v}}\| - s \cdot \sigma| > \frac{C \cdot s \cdot \sigma}{\sqrt{d}}\right) \le \frac{1}{2C^2} + \mathcal{O}\left(\frac{1}{d^2}\right). \tag{14}$$

Further,

$$P\left(\min_i \|\tilde{\boldsymbol{v}}\| >= s \cdot \sigma \cdot \left(1 - \frac{C}{\sqrt{d}}\right)\right) \ge \left(1 - \frac{1}{2C^2} + \mathcal{O}\left(\frac{1}{d^2}\right)\right)^m = 1 - \mathcal{O}\left(\frac{1}{C^2}\right). \tag{15}$$

Combining Equation (13) and Equation (15), we have: with probability at least $\left(1 - \mathcal{O}\left(\frac{1}{C^2}\right)\right) \cdot \left(1 - \mathcal{O}\left(\frac{1}{C^2}\right)\right) = 1 - \mathcal{O}\left(\frac{1}{C^2}\right)$, every two distinct rows has absolute cosine similarity at most $\frac{C}{\sqrt{d}}$, and each row has norm at least $s \cdot \sigma \cdot \left(1 - \frac{C}{\sqrt{d}}\right)$.

When every two distinct rows of $\tilde{\boldsymbol{V}}$ has cosine similarity of $\frac{C}{\sqrt{d}}$, and each row has norm of $s \cdot \sigma \cdot \left(1 - \frac{C}{\sqrt{d}}\right)$, then its nuclear norm is:

$$s\sigma\left(1 - \frac{C}{\sqrt{d}}\right)\left(\sqrt{1 + (m-1)\frac{C}{\sqrt{d}}} + (m-1)\sqrt{1 - \frac{C}{\sqrt{d}}}\right)$$

$$= s\sigma\left(1 - \frac{C}{\sqrt{d}}\right)\left(1 + \frac{1}{2}(m-1)\frac{C}{\sqrt{d}} + \mathcal{O}\left((m-1)^2\frac{C^2}{d}\right) + (m-1)\left(1 - \frac{C}{2\sqrt{d}} + \mathcal{O}\left(\frac{C^2}{d}\right)\right)\right)$$

$$= s\sigma\left(1 - \frac{C}{\sqrt{d}}\right)\left(m + \mathcal{O}\left(\frac{C^2}{d}\right)\right)$$

$$= s \cdot \sigma \cdot m - \mathcal{O}\left(\frac{C}{\sqrt{d}}\right).$$

Therefore,

$$\mathbb{E}[\|\tilde{\boldsymbol{V}}\|_*] \ge \left(1 - \mathcal{O}\left(\frac{1}{C^2}\right)\right) \cdot \left(s \cdot \sigma \cdot m - \mathcal{O}\left(\frac{C}{\sqrt{d}}\right)\right) = s \cdot \sigma \cdot m - \mathcal{O}\left(\frac{C}{\sqrt{d}}\right) - \mathcal{O}\left(\frac{1}{C^2}\right). \tag{16}$$

Combining Equation (12) and Equation (16), we obtain:

$$\mathbb{E}[\|\tilde{\boldsymbol{V}}\|_*] = \mathcal{O}\left(s \cdot \sigma \cdot m\right). \tag{17}$$

**Expected nuclear norm for $m \gg d$.**

When $m \gg d$, $\tilde{V}$ is full rank, with $d$ singular values approximately uniformly distributed.

The singular values of $\tilde{V}$, denoted as $\lambda_i$, for $\quad i = 1, \ldots, d$, are the square roots of the eigenvalues of $\tilde{V}^T \tilde{V}$.

Since the trace of $\tilde{V}^T \tilde{V}$ is the sum of all its eigenvalues, we have:

$$\sum_{i=1}^{d} \lambda_i^2 = \text{Tr}(\tilde{V}^T \tilde{V}) = \sum_{j=1}^{m} \|\tilde{v}_j\|^2. \tag{18}$$

Therefore,

$$\sum_{i=1}^{d} \lambda_i \leq d \cdot \sqrt{\frac{\sum_{i=1}^{d} \lambda_i^2}{d}} = \sqrt{d \cdot \sum_{j=1}^{m} \|\tilde{v}_j\|^2}. \tag{19}$$

Since $\frac{\sqrt{d \cdot \sum_{j=1}^{m} \|\tilde{v}_j\|^2}}{s\sigma}$ follows a chi distribution with $md$ degrees of freedom, *i.e.* $\mathcal{X}_{md}$,

$$\mathbb{E}[\sqrt{d \cdot \sum_{j=1}^{m} \|\tilde{v}_j\|^2}] = s \cdot \sigma \cdot \sqrt{md} - \mathcal{O}\left(\frac{1}{\sqrt{md}}\right). \tag{20}$$

Thus,

$$\mathbb{E}[\|\tilde{V}\|_*] = \mathbb{E}[\sum_{i=1}^{d} \lambda_i] \leq s \cdot \sigma \cdot \sqrt{md} - \mathcal{O}\left(\frac{1}{\sqrt{md}}\right). \tag{21}$$

On the other hand, Rudelson & Vershynin (2009) show that:

$$P\left(\min_i \frac{\sqrt{d}\lambda_i}{s\sigma} \leq \sqrt{m} - \sqrt{d} - t\right) \leq e^{-\frac{t^2}{2}}, \quad t > 0. \tag{22}$$

Let $t = \sqrt{2 \log C}$, where $1 \ll C \ll e^m$. Then, with probability at least $1 - \frac{1}{C}$:

$$\|\tilde{V}\|_* \geq d \cdot \frac{s\sigma}{\sqrt{d}}\left(\sqrt{m} - \sqrt{d} - \sqrt{2 \log C}\right) = s \cdot \sigma \left(\sqrt{md} - \mathcal{O}(d)\right). \tag{23}$$

Therefore,

$$\mathbb{E}[\|\tilde{V}\|_*] \geq \left(1 - \frac{1}{C}\right) \cdot s \cdot \sigma \left(\sqrt{md} - \mathcal{O}(d)\right) = s \cdot \sigma \cdot \sqrt{md} - \mathcal{O}(d) - -\mathcal{O}\left(\frac{1}{C}\right). \tag{24}$$

Combining Equation (21) and Equation (24), we obtain:

$$\mathbb{E}[\|\tilde{V}\|_*] = \mathcal{O}\left(s \cdot \sigma \cdot \sqrt{md}\right). \tag{25}$$

**Correction for non-centered $V$.**

Finally, for the non-centered matrix $V$, we have $\|V\|_* \leq \|\tilde{V}\|_* + \sqrt{m}\|\frac{s\mathbf{1}_d^T}{\sqrt{d}}\|_*$; and $\|V\|_* \geq \|\tilde{V}\|_*$, $\|V\|_* \geq \sqrt{m}\|\frac{s\mathbf{1}_d^T}{\sqrt{d}}\|_*$. When $\sigma$ is large, $\|V\|_*$ is dominated by $\|\tilde{V}\|_*$; whereas when $\sigma$ is small, $\|V\|_*$ is dominated by $\sqrt{m}\|\frac{s\mathbf{1}_d^T}{\sqrt{d}}\|_*$.

We add a correction term of $\mathcal{O}\left(s \cdot \sqrt{m}\right)$ to $\|\tilde{V}\|_*$, yielding:

$$\mathbb{E}[\|V\|_*] = \begin{cases} s \cdot (\sigma \cdot \mathcal{O}(m) + \mathcal{O}(\sqrt{m})), & \text{if } m \ll d, \\ s \cdot \left(\sigma \cdot \mathcal{O}\left(\sqrt{md}\right) + \mathcal{O}(\sqrt{m})\right), & \text{if } m \gg d. \end{cases}$$

$\square$

## A.2 PROOF FOR THEOREM 3.2

*Proof.* The OLE loss is based on the difference between intra-class compactness and inter-class separation. For each class with $N$ augmentations, the nuclear norm of the intra-class representations follows the behavior for $m \ll d$:

$$\mathbb{E}[\|\boldsymbol{V}_{\text{intra}}\|_*] = s \cdot \left( \sigma \mathcal{O}(N) + \mathcal{O}\left( \sqrt{N} \right) \right).$$

For $B$ classes, we sum over the nuclear norms:

$$\mathcal{L}_{\text{intra}} = s \cdot B \left( \sigma \mathcal{O}(N) + \mathcal{O}\left( \sqrt{N} \right) \right).$$

For inter-class separation across all $BN$ samples, the nuclear norm follows the behavior for $m \gg d$:

$$\mathbb{E}[\|\boldsymbol{V}_{\text{all}}\|_*] = s \cdot \left( \sigma \mathcal{O}\left( \sqrt{BNd} \right) + \mathcal{O}\left( \sqrt{BN} \right) \right).$$

Thus, the total OLE loss is:

$$\begin{aligned}
\mathcal{L}_{OLE} &= s \cdot B \left( \sigma \mathcal{O}(N) + \mathcal{O}\left( \sqrt{N} \right) \right) - s \cdot \left( \sigma \mathcal{O}\left( \sqrt{BNd} \right) + \mathcal{O}\left( \sqrt{BN} \right) \right) \\
&= s\sqrt{BN} \left( \sigma \cdot \left( \mathcal{O}\left( \sqrt{BN} \right) - \mathcal{O}\left( \sqrt{d} \right) \right) + \mathcal{O}\left( \sqrt{B} \right) - 1 \right) \\
&= s\sqrt{BN} \left( \sigma \cdot \mathcal{O}\left( \sqrt{BN} \right) + \mathcal{O}\left( \sqrt{B} \right) \right).
\end{aligned}$$

$\square$

## A.3 ANALYSIS OF RANDOM GAUSSIAN MATRIX

To analyze the structure of the matrix $\boldsymbol{V}$ in Lemma 3.1, we start by examining the length of the rows $\boldsymbol{v}_i$. Each row of $\boldsymbol{V}$ is sampled from $\mathcal{N}\left( \frac{s\mathbf{1}_d}{\sqrt{d}}, \frac{s^2\sigma^2 \boldsymbol{I}_d}{d} \right)$, so we first compute the expected norm of $\boldsymbol{v}_i$.

**Expected length of each vector:**

The squared norm of a row $\boldsymbol{v}_i$ is:

$$\mathbb{E}[\|\boldsymbol{v}_i\|^2] = \mathbb{E}\left[ \left\| \frac{s}{\sqrt{d}}\mathbf{1}_d + \tilde{\boldsymbol{v}}_i \right\|^2 \right].$$

Expanding this, we get:

$$\mathbb{E}[\|\boldsymbol{v}_i\|^2] = s^2 + s^2\sigma^2,$$

where $\mathbb{E}[\|\tilde{\boldsymbol{v}}_i\|^2] = s^2\sigma^2$ due to the variance of the Gaussian distribution. Thus, the expected norm is:

$$\mathbb{E}[\|\boldsymbol{v}_i\|] = s\sqrt{1 + \sigma^2}.$$

**Cosine similarity between two vectors:**

The cosine similarity between two rows $\boldsymbol{v}_i$ and $\boldsymbol{v}_j$ is:

$$\cos(\theta_{i,j}) = \frac{\langle \boldsymbol{v}_i, \boldsymbol{v}_j \rangle}{\|\boldsymbol{v}_i\| \|\boldsymbol{v}_j\|}.$$

Since $\boldsymbol{v}_i$ and $\boldsymbol{v}_j$ are independent, we have:

$$\mathbb{E}[\langle \boldsymbol{v}_i, \boldsymbol{v}_j \rangle] = s^2,$$

and the expected cosine similarity is:

$$\mathbb{E}[\cos(\theta_{i,j})] = \frac{s^2}{s^2(1 + \sigma^2)} = \frac{1}{1 + \sigma^2}.$$

For large $\sigma$, the cosine similarity approaches zero and $\theta$ approaches $\frac{\pi}{2}$, meaning the vectors are approximately orthogonal; for small $\sigma$, the cosine similarity approaches one and $\theta$ approaches 0, meaning the vectors are nearly identical. Both $\sigma$ and $s$ control the vector length, and are positively correlated.

A.4 DISCUSSION OF NUCLEAR NORM'S UNITARY INVARIANCE

The nuclear norm of a matrix $\mathbf{A}$, denoted as $\|\mathbf{A}\|_*$, is defined as the sum of its singular values. The singular values of a matrix $\mathbf{A}$ are the square roots of the eigenvalues of $\mathbf{A}^\top \mathbf{A}$. Therefore, for any matrix $\mathbf{V} \in \mathbb{R}^{m \times d}$, the nuclear norm is given by:

$$\|\mathbf{V}\|_* = \sum_{i=1}^{r} \sigma_i(\mathbf{V}),$$

where $\sigma_i(\mathbf{V})$ are the singular values of $\mathbf{V}$, and $r$ is the rank of $\mathbf{V}$.

Now, consider the matrix $\mathbf{PV}$, where $\mathbf{P}$ is a diagonal matrix with diagonal elements $\pm 1$, which flips the signs of the rows of $\mathbf{V}$. To compute the nuclear norm $\|\mathbf{PV}\|_*$, we need to determine its singular values. The singular values are the square roots of the eigenvalues of $(\mathbf{PV})^\top(\mathbf{PV})$:

$$(\mathbf{PV})^\top(\mathbf{PV}) = \mathbf{V}^\top \mathbf{P}^\top \mathbf{PV}.$$

Since $\mathbf{P}$ is a diagonal matrix with $\pm 1$ entries, we have $\mathbf{P}^\top \mathbf{P} = \mathbf{I}_m$, where $\mathbf{I}_m$ is the identity matrix of size $m$. Thus, the expression simplifies to:

$$(\mathbf{PV})^\top(\mathbf{PV}) = \mathbf{V}^\top \mathbf{V}.$$

This shows that the matrix $\mathbf{PV}$ has the same singular values as $\mathbf{V}$, because the eigenvalues of $\mathbf{V}^\top \mathbf{V}$ are unchanged by the multiplication with $\mathbf{P}$.

Therefore, the nuclear norm of $\mathbf{PV}$ is equal to the nuclear norm of $\mathbf{V}$:

$$\|\mathbf{PV}\|_* = \|\mathbf{V}\|_*.$$

A.5 PROOF FOR THEOREM 4.1

*Proof.* The nuclear norm of $V$ is defined as the sum of the singular values of $V$, denoted $\sigma_1, \sigma_2, \ldots, \sigma_N$, so that:

$$\|V\|_* = \sum_{i=1}^{N} \sigma_i.$$

We will use the properties of the singular values to establish bounds on $\|V\|_*$.

First, we know that the sum of the squared singular values is equal to the Frobenius norm of $V$, which is:

$$\sum_{i=1}^{N} \sigma_i^2 = \|V\|_F^2 = N.$$

This is because each column of $V$ is a unit vector, so the total squared length of the matrix is $N$.

We know that the square of singular values of $V$ are eigenvalues of $V^T V$, then we have

$$\sum_{i=1}^{N} \sigma_i^4 = \|V^T V\|_F^2 = N^2 \cdot \overline{\cos^2(\theta)}.$$

Conbining these, we have

$$\begin{cases} \|V\|_* &= \sum_{i=1}^{N} \sigma_i \\ N &= \sum_{i=1}^{N} \sigma_i^2 \\ N^2 \cdot \overline{\cos^2(\theta)} &= \sum_{i=1}^{N} \sigma_i^4 \end{cases}$$

Since $f_1(x) = \sqrt{x}$ is a concave function, and $f_2(x) = x^2$ is a convex function; $\|V\|_* = \sum_{i=1}^{N} f_1(\sigma_i^2)$ is negatively related to $N^2 \cdot \overline{\cos^2(\theta)} = \sum_{i=1}^{N} f_2(\sigma_i^2)$ with fixed $\sum_{i=1}^{N} \sigma_i^2$.

To find the lower and upper bounds of $\|V\|_*$. We need to get the max and min value of $\overline{\cos^2(\theta)}$.

**Minimization of $\overline{\cos^2(\theta)}$:** Using the Cauchy-Schwarz inequality, we have the following chain of inequalities for $\cos(\theta_{i,j})$ where $i \neq j$:

$$\overline{\cos^2(\theta_{i,j})} \geq \overline{|\cos(\theta_{i,j})|}^2 \geq \overline{\cos(\theta_{i,j})}^2.$$

Since $\cos(\theta_{i,i}) = 1$ for all $i$, we focus on the off-diagonal elements $\cos(\theta_{i,j})$ for $i \neq j$. The value of $\overline{\cos^2(\theta)}$ is minimized for a fixed $\overline{\cos(\theta)}$ when all the off-diagonal cosine similarities $\cos(\theta_{i,j})$ are identical, i.e., when the vectors form an equally spaced configuration in high-dimensional space.

For such a configuration, the matrix $\boldsymbol{V}^T\boldsymbol{V}$ has one eigenvalue corresponding to the eigenvector $\mathbf{1} = (1, 1, \ldots, 1)^\top$, with eigenvalue $\sqrt{N\overline{\cos(\theta)}}$. The remaining $N - 1$ eigenvalues have the same value $\frac{\sqrt{N - N\overline{\cos(\theta)}}}{N-1}$ and correspond to eigenvectors orthogonal to $\mathbf{1}$.

Therefore,

$$\|\boldsymbol{V}\|_* \leq \sqrt{N} \cdot \sqrt{\overline{\cos(\theta)}} + \sqrt{N(N-1)} \cdot \sqrt{1 - \overline{\cos(\theta)}}.$$

**Maximization of $\overline{\cos^2(\theta)}$:** We note that $\cos^2(\theta_{i,j}) \leq |\cos(\theta_{i,j})|$. Equality holds when $|\cos(\theta_{i,j})|$ is either 1 or 0, meaning that the vectors are either perfectly aligned, completely anti-aligned, or orthogonal to each other. This scenario occurs when the vectors can be grouped into $G$ distinct groups, such that within each group, the vectors are either aligned or anti-aligned, while vectors from different groups are orthogonal.

Using this setup, the matrix $\boldsymbol{V}^T\boldsymbol{V}$ will have $G$ non-zero singular values, each corresponding to a group of vectors. To maximize the nuclear norm, we apply the Cauchy-Schwarz inequality to obtain:

$$\|\boldsymbol{V}\|_* \geq \sqrt{GN},$$

where $G$ represents the number of groups.

**Minimizing $G$:** We now need to find the minimum possible value of $G$. The minimum value of $G$ occurs when the vectors are partitioned into the fewest number of groups, while ensuring that the vectors within each group are either aligned or anti-aligned, and vectors between different groups are orthogonal. The smallest $G$ is given by:

$$\frac{N^2 \cdot \overline{\cos^2(\theta)}}{G} \geq \left(\frac{N}{G}\right)^2 \implies G \geq \frac{1}{|\cos(\theta)|}.$$

Substituting this into the earlier inequality for the nuclear norm, we obtain:

$$\|\boldsymbol{V}\|_* \geq \sqrt{\frac{N}{|\cos(\theta)|}}.$$

Combining the results from the maximum and minimum cases, we obtain the desired bounds for the nuclear norm:

$$\sqrt{\frac{N}{|\cos(\theta)|}} \leq \|\boldsymbol{V}\|_* \leq \sqrt{N} \cdot \sqrt{\overline{\cos(\theta)}} + \sqrt{N(N-1)} \cdot \sqrt{1 - \overline{\cos(\theta)}}.$$

$\square$

### A.6 Proof for Theorem 4.2

*Proof.* Let $\boldsymbol{V} \in \mathbb{R}^{d \times N}$ be a matrix where the $i$-th column is the unit vector $\boldsymbol{v}_i \in \mathbb{R}^d$, and define $\bar{\boldsymbol{v}} := \frac{1}{N} \sum_{i=1}^{N} \boldsymbol{v}_i$ as the mean vector. The deviation matrix is defined as $\tilde{\boldsymbol{V}} = \boldsymbol{V} - \bar{\boldsymbol{v}}\mathbf{1}^\top$, where $\mathbf{1} \in \mathbb{R}^N$ is a vector of ones.

The Frobenius norm of the deviation matrix is:

$$\|\tilde{\boldsymbol{V}}\|_F^2 = \sum_{i=1}^{N} \|\boldsymbol{v}_i - \bar{\boldsymbol{v}}\|^2.$$

For each vector $\boldsymbol{v}_i$, we can expand $\|\boldsymbol{v}_i - \bar{\boldsymbol{v}}\|^2$ as:

$$\|\boldsymbol{v}_i - \bar{\boldsymbol{v}}\|^2 = 1 - 2\boldsymbol{v}_i^\top \bar{\boldsymbol{v}} + \|\bar{\boldsymbol{v}}\|^2.$$

We calculate $\|\bar{\boldsymbol{v}}\|^2$ as:

$$\|\bar{\boldsymbol{v}}\|^2 = \frac{1}{N^2} \sum_{i=1}^{N} \sum_{j=1}^{N} \boldsymbol{v}_i^\top \boldsymbol{v}_j = \frac{1}{N^2} \sum_{i=1}^{N} \sum_{j=1}^{N} \cos(\theta_{i,j}),$$

where $\theta_{i,j}$ is the angle between vectors $\boldsymbol{v}_i$ and $\boldsymbol{v}_j$. Thus, the Frobenius norm of the deviation matrix is:

$$\|\tilde{\boldsymbol{V}}\|_F^2 = N \left( 1 - \overline{\cos(\theta)} \right),$$

where $\overline{\cos(\theta)}$ is the average cosine similarity.

The nuclear norm is the sum of the singular values of the matrix. The sum of the squared singular values equals the Frobenius norm, $\|\tilde{\boldsymbol{V}}\|_F^2 = N(1 - \overline{\cos(\theta)})$.

The nuclear norm is bounded between two extremes. The lower bound occurs when the vectors can be grouped into two perfectly aligned sets. In this case, the deviation vectors are either identical or the opposite. The deviation from the mean will be minimized, and the nuclear norm satisfies:

$$\|\tilde{\boldsymbol{V}}\|_* \geq \sqrt{N} \cdot \sqrt{1 - \overline{\cos(\theta)}}.$$

The upper bound is attained when the angles between all vectors are identical. In this configuration, the deviation vectors are orthogonal to each other, yielding:

$$\|\tilde{\boldsymbol{V}}\|_* \leq \sqrt{N(N-1)} \cdot \sqrt{1 - \overline{\cos(\theta)}}.$$

Therefore, the nuclear norm of the deviation matrix is bounded by:

$$\sqrt{N} \cdot \sqrt{1 - \overline{\cos(\theta)}} \leq \|\tilde{\boldsymbol{V}}\|_* \leq \sqrt{N(N-1)} \cdot \sqrt{1 - \overline{\cos(\theta)}}.$$

$\square$

## B    LIMITATIONS

While this study introduces SSOLE as a robust and efficient training objective for Multi-view Self-Supervised Learning (MV-SSL), integrating Orthogonal Low-rank Embedding (OLE) concepts into SSL, it does have certain limitations. Our experiments validate SSOLE's performance using ResNet-18 and ResNet-50 on ImageNet100 and the full ImageNet(1K) dataset. However, the potential of SSOLE with more complex architectures, such as wide ResNet-50, Vision Transformers (ViT), and Swin Transformers, remains unexplored. These architectures have shown promise in SSL, achieving state-of-the-art results. Future work could explore the applicability of SSOLE with these advanced architectures. Additionally, investigating SSOLE as a complementary regularization technique alongside other SSL methods presents another promising avenue for future research.

## C    IMPLEMENTATION DETAILS

This section outlines the configurations for our experiments in ablation studies, focusing on InfoNCE-M and SSOLE methods.

### C.1    INCORPORATION OF MULTIPLE CROPS

The multiple crops technique, introduced in SwAV Caron et al. (2020), draws inspiration from discussions in Wang et al. (2022) on handling outlier views and balancing low-rank and high-rank constraints. In SSL, the diversity of views during training has a significant impact on the effectiveness of models. As highlighted in Wang et al. (2022), incorporating high-variance views such as small crops requires careful consideration because they can deviate from the principal components of larger views. These deviations challenge the low-rank assumptions central to SSL, necessitating a balanced approach in multi-view learning.

To address this, we propose a two-step approach to low-rank and high-rank enforcement for large and small crops. Our strategy ensures that gradients from small crops do not interfere with those from large crops, which represent the primary views used for low-rank enforcement. The steps are detailed as follows:

Let $\mathbf{Z}_L \in \mathbb{R}^{B \times N^L \times d}$ represent the feature matrix of the large crops, where $N^L$ is the number of large crops, and $d$ is the feature dimension. First, we compute the deviation matrix $\tilde{\mathbf{Z}}_{L,b,:}$ by subtracting the mean of the large crops' feature vectors:

$$\tilde{\mathbf{Z}}_{L,b,:} = \mathbf{Z}_{L,b,:} - \text{sg}(\frac{1}{N^L} \sum_{n=1}^{N^L} \mathbf{Z}_{L,b,n}),$$

where $\text{sg}(\cdot)$ denotes the stop-gradient operation. The low-rank loss for the large crops is then applied to this deviation matrix as:

$$\mathcal{L}_{intra}^{(L)} = h_1 \left( \|\tilde{\mathbf{Z}}_{L,b,:}\|_*, N^L \right).$$

Let $\mathbf{Z}_{\text{all}} \in \mathbb{R}^{B \times N^{\text{all}} \times d}$ represent the feature matrix for all crops, where $N^{\text{all}}$ includes both large (L) and small (S) crops. To account for the variance introduced by small crops, we calculate the deviation matrix for all crops, but the mean is replaced by the mean of the large crops:

$$\tilde{\mathbf{Z}}_{\text{all},b,:} = \mathbf{Z}_{\text{all},b,:} - \text{sg}(\frac{1}{N^L} \sum_{n=1}^{N^L} \mathbf{Z}_{L,b,n}).$$

To further prevent the gradients of small crops from affecting the large crops, we stop the gradient flow on the large crops in this step. The low-rank loss for all crops is then computed as:

$$\mathcal{L}_{intra}^{(All)} = h_1 \left( \|\tilde{\mathbf{Z}}_{\text{all},b,:}\|_*, N^{\text{all}} \right).$$

Then the low-rank loss for small crops is then computed as:

$$\mathcal{L}_{intra}^{(S)} = 2 * \mathcal{L}_{intra}^{(All)} - \text{sg}(\mathcal{L}_{intra}^{(L)}).$$

For high-rank enforcement, we perform the same operation on all crops, including both large and small views. Let $\mathbf{Z}_{:,n}$ represent the feature matrix of the $n$-th view. The high-rank loss is computed as:

$$\mathcal{L}_{inter}^L = h_2 \left( \|\mathbf{Z}_{L:,n}\|_*, B \right),$$
$$\mathcal{L}_{inter}^S = h_2 \left( \|\mathbf{Z}_{S:,n}\|_*, B \right).$$

To adjust the influence of the small crops, we introduce a weighting factor $\beta$. The low-rank and high-rank losses for the small crops are scaled by $\beta$, ensuring that the small crops' contribution is balanced against that of the large crops:

$$\mathcal{L}_{intra} = \mathcal{L}_{intra}^{(L)} + \beta * \mathcal{L}_{intra}^{(S)},$$
$$\mathcal{L}_{inter} = \mathcal{L}_{inter}^{(L)} + \beta * \mathcal{L}_{inter}^{(S)}.$$

This approach ensures that diverse crops are incorporated effectively while preserving the benefits of low-rank and high-rank enforcement. The factor $\beta$ allows for fine-tuning of the influence of small crops, optimizing the balance between view diversity and stability in SSL.

## C.2 PRETRAINING

**ImageNet100 Experiments:** We employ ResNet-18 as the backbone ($f_\theta$) with a three-layer MLP (4096-$d$ hidden layer with ReLU, followed by normalization) as a projector, yielding a final embedding dimension of $d = 4096$. Training utilizes a batch size of $B = 128$ across 4 GPUs, SGD optimizer with a base learning rate of $lr = 2.0$, and a cosine decay to 0.002. $\lambda = 0.7$. The experiment uses $N_L = 4$ full views and $N_S = 4$ small views with $\beta = 0.6$.

**Full ImageNet (1K) Experiments:** For the full ImageNet dataset, ResNet-50 is used as the backbone with an enhanced three-layer MLP (8192-$d$ with ReLU and normalization) in the projector, leading

to an embedding size of $d = 8192$. The batch size is set at $B = 256$, evenly distributed over 8 GPUs. The SGD optimizer is used with a base learning rate of $lr = 1.0$, decaying to $0.001$ following a cosine rule. $\lambda = 0.7$. The experiment uses $N_L = 4$ full views and $N_S = 4$ small views with $\beta = 0.6$.

We adopt the "multi-crop" data augmentation strategy from SwAV, an exemplary multi-view training algorithm. This method enriches the diversity of input data, crucial for effective multi-view self-supervised learning. During each iteration of training, we generate a combination of views:

- *Full Views*: We create $N_L$ full views for each image, each of size $224 \times 224$ pixels. The scale factor for these views varies within the range of $[0.14, 1.0]$, ensuring a wide representation of the original images.

- *Small Views*: Alongside full views, $N_S$ small views of size $96 \times 96$ pixels are generated, with a scale factor ranging from $[0.05, 0.33]$. These smaller views focus on different segments of the images, introducing further variance.

Each generated view undergoes a series of augmentation techniques to enhance model robustness:

- *Random Horizontal Flip*: Applied with a probability of 0.5 to introduce horizontal variability.

- *Color Distortion*: This includes color jittering (brightness, contrast, saturation, and hue adjustments with respective strengths of 0.8, 0.8, 0.8, and 0.2) applied with a probability of 0.8, and color dropping (conversion to grayscale) with a probability of 0.2.

- *Gaussian Blur*: Each view is subjected to Gaussian blur, having a standard deviation in the range of $[0.1, 2.0]$, to simulate focus variability.

- *Random Solarization*: Applied with a probability of 0.2 to further diversify the visual input.

### C.3 LINEAR PROBING

Linear probing evaluates the representational quality of our SSOLE model. We outline distinct training protocols for ImageNet100 and the full ImageNet dataset.

**Training Protocols:**

- *ImageNet100*:
  - Batch Size: $B = 256$.
  - Optimizer: SGD with a momentum of 0.9, no weight decay.
  - Learning Rate: Base rate $lr = 0.1$, cosine schedule over 100 epochs.
- *Full ImageNet*:
  - Batch Size: $B = 2048$.
  - Optimizer: SGD, momentum 0.9, no weight decay.
  - Learning Rate: Starting at $lr = 0.6$, reduced by 0.3 every 20 epochs, across 100 epochs.

**Data Augmentation:**

- *Training Images*: Random cropping and resizing to $224 \times 224$, plus random horizontal flips (probability 0.5).
- *Test Images*: Resize to $256 \times 256$, then center crop to $224 \times 224$.

**Implementation Note:** For the full ImageNet, we implemented SwAV based on the available code and compared our results with those reported in Wang et al. (2022) (referenced in Table 4).

### C.4 SEMI-SUPERVISED LEARNING ON IMAGENET

We conduct fine-tuning experiments with limited labeled data, specifically 1% and 10% subsets of the ImageNet dataset. These subsets are the same as those used in Chen et al. (2020a).

For the fine-tuning process, we employ an SGD optimizer with a batch size of $B = 256$ and a momentum of 0.9, without any weight decay. The fine-tuning is carried out for 20 epochs for both the 1% and 10% labeled datasets.

Following the learning rate scaling strategy from SwAV, we set different learning rates for the linear layers and the backbone network weights. Specifically, the linear layers' learning rates are scaled up by 250 times and 20 times for the 1% and 10% tasks, respectively. We determined the optimal base learning rates for the linear layers to be 5.0 for the 1% task and 0.2 for the 10% task after conducting a search in the range of 0.01 to 10. These learning rates are then reduced by a factor of 0.2 at the 12th and 16th epochs during the training period.

### C.5 TRANSFER LEARNING FROM IMAGENET TO OTHER DATASETS

For transfer learning tasks, we maintain consistent image pre-processing protocols during linear classifier training and testing, as used in the linear evaluation on ImageNet. However, for CIFAR10 and CIFAR100, we adjust the image size to $224 \times 224$ and apply random horizontal flipping to training images with a probability of 0.5.

In these experiments, our focus is on training linear classifiers on top of the frozen feature representations extracted from the pre-trained network. We employ an SGD optimizer with a batch size of $B = 256$, momentum of 0.9, and no weight decay. To determine the optimal base learning rate for each algorithm, we conduct a search across seven logarithmically-spaced values ranging from 0.1 to 100. Once the optimal learning rate is identified for each algorithm, we apply a cosine decay rule for its annealing.

For the CIFAR10 and CIFAR100 datasets, the linear classifiers are trained for a total of 30,000 iterations. In contrast, for the Aircraft, DTD, and Flowers datasets, we limit the training to 5,000 iterations.

We report top-1 accuracies (%) achieved by each method on CIFAR10, CIFAR100, and DTD datasets; and *mean per class* on Aircraft and Flowers, following practices in Wang et al. (2022); Grill et al. (2020).

