# OpenReview forum: "SSOLE: Rethinking Orthogonal Low-rank Embedding for Self-Supervised Learning"
_ICLR.cc/2025/Conference — ICLR 2025 Poster_

### Official Review · Reviewer_dJ94 · 2024-10-30

**Soundness:** 3
**Presentation:** 3
**Contribution:** 3
**Rating:** 8
**Confidence:** 4

**Summary:**

This paper proposes a self-supervised orthogonal low-rank embedding (SSOLE), which integrates OLE into the SSL paradigm. It addresses two challenges in applying OLE to SSL: the difficulty of enforcing orthogonality in the presence of an infinite number of classes, and the nuclear norm’s inability to distinguish between positive and negative correlations. By decoupling low-rank and high-rank enforcement and applying low-rank constraints to feature deviations, SSOL adapts OLE for self-supervised and other tasks.

**Strengths:**

1.	This paper is well-written with clear motivations.
2.	It is technically sound with comprehensive theoretical analysis.
3.	Experimental results demonstrate the effectiveness of the method.

**Weaknesses:**

1.	The parameter $\lambda$ controls the balance between intra-class compactness and inter-class separability enforcement. It will be better to analyze its influence to the final performance.

2.	The authors enforce intra-class low-rank property via deviation matrix instead of original feature matrix, it is also suggested to investigate its effectiveness by ablation study.

**Questions:**

See weaknesses.

---

> ### Author Response · Authors · 2024-11-22
> **Response to Reviewer dJ94**
>
> We sincerely thank Reviewer dJ94 for their thoughtful review and positive evaluation of our work. We are particularly encouraged by the recognition of our clear motivations, sound theoretical analysis, and effective experimental results. Below, we address the specific concerns raised regarding parameter sensitivity and the use of the deviation matrix, and we provide additional clarifications.
>
> # 1. Parameter $\lambda$ and Its Influence
> >The parameter $\lambda$ controls the balance between intra-class compactness and inter-class separability enforcement. It will be better to analyze its influence on the final performance.
>
> We completely agree that studying the influence of $\lambda$ is critical for understanding the behavior of SSOLE. In response, we conducted an additional ablation study by varying $\lambda$ across a broad range of values on the ImageNet-100 dataset (5 views, 100 epochs, linear probing Top-1 accuracy):
>
> | λ    | 0.1  | 0.2  | 0.3  | 0.4  | 0.5  | 0.6  | 0.7  | 0.8  | 0.9  | 1.0  | 1.1  | 1.2  | 1.3  | 1.4  | 1.5  | 1.6  | 1.7  | 1.8  | 1.9  | 2.0  | 2.5  | 3.0  |
> |------|------|------|------|------|------|------|------|------|------|------|------|------|------|------|------|------|------|------|------|------|------|------|
> | Acc. | 72.1 | 73.9 | 76.5 | 77.5 | 78.4 | 78.5 | 78.5 | 78.4 | 78.5 | 78.0 | 78.3 | 78.6 | 78.3 | 78.3 | 78.1 | 78.3 | 78.4 | 78.3 | 78.4 | 78.0 | 77.3 | 77.3 |
>
> The results demonstrate:
> - For smaller values of $\lambda \in [0.1, 0.4]$, the model prioritizes intra-class compactness (low-rank constraints), which can lead to under-penalized inter-class overlap, slightly degrading performance.
> - For larger values of $\lambda \in [2.0,3.0]$, the model emphasizes inter-class separability (high-rank constraints), sometimes at the expense of intra-class consistency, resulting in over-dispersed features.
> - The optimal range of  $\lambda$ is $[0.5, 1.9]$, where the model balances compactness and separability, achieving its best performance. Notably, the model shows high robustness to $\lambda$ within this range, indicating a stable balance between objectives.
>
> We have included this analysis in the revised draft to better illustrate the influence of $\lambda$ in Section 6.1.1.
>
> # 2. Effectiveness of the Deviation Matrix
> >The authors enforce intra-class low-rank property via the deviation matrix instead of the original feature matrix. It is suggested to investigate its effectiveness through ablation studies.
>
> Thank you for pointing this out. We agree that understanding the role of the deviation matrix is important. To clarify, we have already conducted an ablation study in the paper, comparing the use of the deviation matrix against the original feature matrix for enforcing low-rank constraints. This comparison is detailed in Table 1 of our submission.
>
> The results demonstrate that directly applying low-rank constraints to the original feature matrix, (corresponding to the training objective $\mathcal{L}_{\rm{OLE}}$ + “normalization” + “loss decoupling”), leads to significant training instability:
> - When using the original feature matrix, the model exhibited high sensitivity to the hyperparameter $\lambda$. Specifically:
>   - For $\lambda \leq 2.10$, training collapsed into trivial constant solutions.
>   - For $\lambda \geq 2.20$, training produced random embeddings.
>   - Temporary stabilization occurred at $\lambda = 2.15$, but even in this case, training was unstable and prone to SVD-related errors.
> - In contrast, enforcing low-rank constraints on the deviation matrix resolved these issues:
>   - The deviation matrix naturally centers feature vectors by subtracting the mean, which ensures that optimization focuses on deviations rather than absolute magnitudes.
>   - This approach significantly improved stability, allowing consistent convergence across a wider range of $\lambda$, as demonstrated in the above ablation study.
>   - Furthermore, the deviation matrix showed enhanced robustness to $\lambda$, as evidenced by the results across [0.5, 1.9], where SSOLE outperformed baseline methods.
>
> These findings highlight the importance of the deviation matrix in adapting OLE to SSL. To ensure this is clear to readers, we have revised Section 6.1.1 to emphasize this analysis and its implications.
>
> We hope this response addresses the reviewer’s concerns comprehensively and demonstrates the improvements we have made to strengthen the paper. Thank you for your constructive feedback.

---

> > ### Comment · Reviewer_dJ94 · 2024-11-26
> >
> > Thank you for your detailed responses. My concerns are well addressed, and I maintain my scores as Accept.

---

> > > ### Author Response · Authors · 2024-11-26
> > > **Thanks**
> > >
> > > Thank you for your thorough and thoughtful review. We appreciate your time and the valuable contributions you have made to our paper.
> > >
> > > Warm Regards,
> > >
> > > Authors

---

### Official Review · Reviewer_3eqn · 2024-11-03

**Soundness:** 3
**Presentation:** 3
**Contribution:** 3
**Rating:** 6
**Confidence:** 3

**Summary:**

The paper points out that traditional Orthogonal Low-Rank Embedding (OLE) methods face significant challenges in self-supervised learning (SSL), mainly due to representational collapse caused by an excessively large number of classes, and the difficulty of distinguishing between positively and negatively correlated features under low-rank constraints. To address these issues, SSOLE decouples the low-rank and high-rank constraints and applies low-rank constraints to the deviation matrix of features. This approach effectively prevents representational collapse and enhances the ability to differentiate between positive and negative sample pairs. Experimental results demonstrate that SSOLE achieves excellent performance across various SSL benchmarks, showing good scalability and efficiency, especially without requiring large batch sizes and complex architectures.

**Strengths:**

1. Methodological Innovation: The paper introduces a low-rank bias matrix and a decoupled constraint mechanism based on the integration of OLE and SSL, addressing the issue of representation collapse that traditional methods struggle to resolve in unsupervised scenarios.

2. Theoretical and Experimental Support: The effectiveness of the SSOLE framework is supported by both theoretical analysis and experimental validation, demonstrating strong performance across various datasets, particularly in settings with limited computational resources.

3. Broad Applicability: In image classification tasks, SSOLE shows good generalization ability, adapting to different datasets and further enhancing the robustness of feature representation.

**Weaknesses:**

1. Some proofs are omitted and not specific enough, such as

 i) Why is the probability of $|cos(\theta_{ij})|>1/d$ is larger than $1/d$? By what, Chebyshev inequality?

 ii) Some statements are as follows: "Since the rows of  $\tilde{V}$ are nearly orthogonal, the nuclear norm is dominated by the sum of the row norms." Why can it hold? The readers need more detailed explanations.

2. Some proofs are unnecessary. THEOREM 3.3 states that the nuclear norm is unitarily invariant, a property that is very common in linear algebra textbooks.

3. What does $\approx$ mean? Does it mean that the equation holds with high probability, or that the values are close? If so, how close are the values? The paper needs to give a clear definition.

**Questions:**

See "Weaknesses".

---

> ### Author Response · Authors · 2024-11-22
> **Response to Reviewer 3eqn**
>
> We thank Reviewer 3eqn for the detailed and thoughtful feedback on our work. We are pleased that the reviewer found our methodological innovation, theoretical analysis, and experimental validation to be strong, and that our framework's scalability and efficiency were acknowledged. Below, we address the specific concerns raised regarding the omitted proofs, the clarity of certain statements, and the necessity of some theorems.
>
> # 1. Clarity of Specific Proofs
> >Why is the probability of $|\cos(\theta_{i,j})| > \frac{1}{d}$ larger than $\frac{1}{d}$? By what method, such as Chebyshev inequality?
>
> Thank you for pointing out the need for additional clarity. There was indeed a typo that caused confusion. In the revised draft, we provide a more rigorous proof in the updated Appendix A.1. Specifically:
> - Using Chebyshev inequality, we show that the probability of $|\cos(\theta_{i,j})| > \frac{C}{\sqrt{d}}$ is bounded by $\frac{1}{C^2}$, leveraging the fact that the cosine similarity between rows, $|\cos(\theta_{i,j})|$, is distributed with mean 0 and variance of $\frac{1}{d}$ for random vectors on a unit sphere. Full details are included in the updated Appendix A.1.
>
> > Why does the statement "Since the rows of $\tilde{\mathbf{V}}$ are nearly orthogonal, the nuclear norm is dominated by the sum of the row norms" hold? The readers need more detailed explanations.
>
> This statement leverages the approximate orthogonality of the rows of $\tilde{\mathbf{V}}$. We provide a formal justification in the updated Appendix A.1:
> - For perfectly orthogonal rows, the singular values of $\tilde{\mathbf{V}}$ are simply the row norms, and thus the nuclear norm is the sum of these norms. When the rows are nearly orthogonal, deviations in cosine similarity (bounded by $\frac{C}{\sqrt{d}}$​) result in small perturbations to this sum. We quantify these deviations in the appendix to show the approximation is valid.
>
>
> # 2. Necessity of Theorem 3.3
> >Some proofs, such as Theorem 3.3 (nuclear norm is unitarily invariant), are unnecessary and are standard properties in linear algebra.
>
> While the unitary invariance of the nuclear norm is a standard result, we include it to connect this property to its specific application in our context of deviation matrices and feature alignment. Removing Theorem 3.3 entirely could create a gap in the logical flow, as readers unfamiliar with linear algebra might struggle to connect this standard result to its specific application in our framework. To balance clarity and brevity, we have relocated the discussion to Appendix A.4 while keeping references in the main text for continuity.
>
> # 3. Clarification of Approximation (" $\approx$ ")
> >What does " $\approx$ " mean? Does it indicate high probability, or that the values are close? If so, how close are the values?
>
> The symbol " $\approx$ " in our paper indicates that the stated equality holds up to small error terms that diminish as d or m grows. For example, in the analysis of cosine similarities between rows of random Gaussian matrices, " $\approx$ " indicates that deviations from expected values are controlled by concentration bounds that diminish as d grows. These bounds are quantified in the appendix to ensure precision.
>
> To avoid ambiguity, we have replaced all occurrences of " $\approx$ " with precise terminology:
> - **High Probability Statements**: When " $\approx$ " signifies a high-probability bound (e.g., with deviation from the mean controlled by concentration inequalities), we explicitly state this.
> - **Numerical Approximation**: When the equality represents an approximation derived from numerical truncation (e.g., singularvalue bounds), we quantify the error bounds.
>
> We appreciate Reviewer 3eqn’s positive evaluation and constructive feedback. To summarize our revisions:
> - We clarified and rigorously proved all theoretical claims, particularly those highlighted in the review.
> - We replaced all occurrences of " $\approx$ " with precise terminology throughout the paper to ensure precision and quantified approximations.
> - We refined the presentation of Theorem 3.3, emphasizing its relevance without redundancy.
>
> We hope this response addresses the reviewer’s concerns comprehensively and demonstrates the improvements we have made to strengthen the paper. Thank you for your constructive feedback.

---

### Official Review · Reviewer_D3sb · 2024-11-03

**Soundness:** 3
**Presentation:** 3
**Contribution:** 3
**Rating:** 8
**Confidence:** 3

**Summary:**

This paper primarily focuses on applying OLE to SSL and propose a novel method that integrates Orthogonal Low-rank Embedding into the Self-Supervised Learning paradigm. The authors mainly addresses two key challenges in applying OLE to SSL: the enforcement of orthogonality with an infinite number of classes and the limitations of the nuclear norm in distinguishing between positive and negative correlations. By decoupling low-rank and high-rank enforcement and applying constraints on feature deviations, SSOLE adapts OLE for self-supervised tasks. The paper demonstrates how SSOLE adapts OLE for self-supervised tasks and showcases its superior performance in various learning scenarios while maintaining computational efficiency.

**Strengths:**

1.The paper offers comprehensive experimentation, strengthening the validity of the presented approach.
2.The method is described in detail, enhancing its reproducibility and understanding.
3.In terms of experiments,  this paper evaluate the adaptability and robustness of the SSOLE framework through transfer
learning to various linear classification tasks and demonstrates its superior performances.

**Weaknesses:**

1.The writing of this paper needs to be improved.
2.Some experiments are not sufficiently thorough, such as when evaluating the performance of the method in semi-supervised learning, the datasets used are somewhat limited, and the experimental results are not particularly striking. If additional experiments on other datasets could be conducted to demonstrate the method's effectiveness, it would be more convincing.

**Questions:**

1.In the case of weakly supervised datasets, such as when the dataset contains noisy labels, does this method have adaptability?
2.Regarding the description in the appendix A.4 that the product of diagonal matrix P and its transpose is the identity matrix, why are all the diagonal elements of the resulting matrix either 1 or -1? Could you provide a more detailed explanation?

---

> ### Author Response · Authors · 2024-11-22
> **Response to Reviewer D3sb (Part 1)**
>
> We appreciate the time and effort Reviewer D3sb has taken to assess our work. We are encouraged by the recognition of our experimental rigor, detailed method description, and strong performance in transfer learning tasks. Below, we address the concerns raised and provide clarifications to ensure our contributions and findings are well understood.
>
> # 1. Writing Clarity
> >The writing of this paper needs to be improved.
>
> We acknowledge the importance of clear writing for effective communication. In response, we have revised the manuscript to address potential ambiguities and enhance clarity. Specifically, we have:
> - Improved the explanation of the relationship between limitations, challenges, and our method in Section 3 to explicitly connect the issues of traditional OLE with the adaptations required for SSL.
> - Refined the technical explanations in the Appendix, to provide more detailed derivations and eliminate potential confusion.
> - Enhanced the discussion of experimental results to better highlight the significance of our contributions.
>
> These revisions improve readability and ensure a more coherent narrative throughout the paper. We hope these updates address the reviewer’s concern about writing quality. If there are specific areas still in need of further clarification, we welcome additional feedback. Having said so, for the published version we plan to go in detail over all the writing; there was no sufficient time during the response period and we wanted to focus on the technical responses first.
>
> # 2. Thoroughness of Experiments
> >Experiments on semi-supervised learning are somewhat limited, and results are not particularly striking. Additional datasets could make the method more convincing.
>
> We appreciate this feedback and agree that demonstrating the adaptability of SSOLE across diverse tasks and datasets strengthens the validity of our approach. We have conducted additional experiments on **Coco Object Detection and Instance Segmentation Transfer Learning**. These tasks assess the adaptability and generalization of SSOLE beyond linear evaluation and semi-supervised learning. Below is a summary of the new results, demonstrating that SSOLE achieves state-of-the-art performance:
>
> | Model           | AP$_{50}$ | AP   | AP$_{75}$ | AP$^{mask}_{50}$ | AP$^\{mask}$ | AP$^{mask}_{75}$ |
> |------------------|-----------|------|-----------|--------------------------|--------------------------|--------------------------|
> | SimCLR          | 57.7      | 37.9 | 40.9      | 54.6                     | 33.3                     | 35.3                     |
> | MoCo v2         | 58.9      | 39.3 | 42.5      | 55.8                     | 34.4                     | 36.5                     |
> | BYOL            | 57.8      | 37.9 | 40.9      | 54.3                     | 33.2                     | 35.0                     |
> | SwAV            | 58.6      | 38.4 | 41.3      | 55.2                     | 33.8                     | 35.9                     |
> | SimSiam         | 59.3      | 39.2 | 42.1      | 56.0                     | 34.4                     | 36.7                     |
> | Barlow Twins    | 59.0      | 39.2 | 42.5      | 56.0                     | 34.3                     | 36.5                     |
> | VICReg          | -         | 40.0 | -         | -                        | -                        | 36.7                     |
> | MEC             | 59.8      | 39.8 | 43.2      | 56.3                     | 34.7                     | 36.8                     |
> | Matrix-SSL      | 60.8      | 41.0 | 44.2      | 57.5                     | 35.6                     | 38.0                     |
> | INTL            | 61.0      | 41.0 | 44.5      | 57.7                     | 35.6                     | 37.8                     |
> | **SSOLE (Ours)**| **61.5**  | **41.3** | **44.8** | **58.0**                | **35.9**                | **38.4**                |
>
> These results demonstrate that SSOLE achieves the best performance in both object detection and instance segmentation, highlighting its robustness and adaptability across tasks. We believe these additional results provide compelling evidence of the method’s generalizability.
> The experiments are included in the revised draft (Section 6.3.2 and Table 5).

---

> ### Author Response · Authors · 2024-11-22
> **Response to Reviewer D3sb (Part 2)**
>
> # 3. Adaptability to Noisy Labels
> >In the case of weakly supervised datasets, such as when the dataset contains noisy labels, does this method have adaptability?
>
> While noisy labels are not the focus of our work, we argue that SSOLE is naturally robust in such scenarios for the following reasons:
> - **Label-Free Pre-training**: SSOLE leverages SSL during pre-training, meaning the representations are learned without any reliance on labels. This ensures that label noise during fine-tuning does not compromise the learned features.
> - **Low-rank Regularization**: The low-rank enforcement aligns representations of the same instance, which can help smooth over noisy supervision when applied to fine-tuning tasks. For instance, outlier labels often result in noisy gradients, but low-rank alignment helps counteract this effect by promoting consistency among features.
>
> These properties suggest that SSOLE can adapt to weakly supervised scenarios. While we have not specifically evaluated noisy datasets in this submission, future work could empirically validate this robustness. We will also consider adding this to the revision if we obtain results on time.
>
> # 4. Clarification on Appendix A.4
> >In Appendix A.4, why are the diagonal elements of $\mathbf{P}$ either 1 or −1? Could you provide a more detailed explanation?
>
> We apologize for the confusion caused by the description in Appendix A.4. The claim in our paper is that if the diagonal elements of $\mathbf{P}$ are either 1 or -1, then:
>
> $\mathbf{P}^T \mathbf{P} = \mathbf{I}$.
>
> To clarify:
> - $\mathbf{P}$ is a diagonal matrix where each diagonal entry is $\pm 1$.
> - When $\mathbf{P}^T$ is multiplied by $\mathbf{P}$, the result is the identity matrix because:
>   - The product of any diagonal element with its transpose is 1 ($(+1)^2 = 1,(-1)^2 = 1$).
>   - All off-diagonal elements are 0 because $\mathbf{P}$ is diagonal.
>
> This means that $\mathbf{P}$ does not alter the eigenvalues of $\mathbf{V}^T\mathbf{V}$, or singular values of $\mathbf{V}$, which is a result of the unitarily invariance property of the nuclear norm.
>
> We have refined the presentation of Theorem 3.3 to improve clarity.
>
> We hope this response addresses the reviewer’s concerns comprehensively and demonstrates the improvements we have made to strengthen the paper. Thank you for your constructive feedback.

---

> ### Comment · Reviewer_D3sb · 2024-11-25
>
> Dear Authors,
>
> I appreciate the thoroughness with which you have addressed the concerns raised. The experiments conducted demonstrate the robustness of your approach and its effectiveness in various scenarios, which adds substantial value to the paper. The supplementary materials and further analyses have provided a clearer understanding of the method's strengths and potential areas for future work.

---

> > ### Author Response · Authors · 2024-11-26
> > **Thanks**
> >
> > Thank you for your thorough and thoughtful review. We appreciate your time and the valuable contributions you have made to our paper.
> >
> > Warm Regards,
> >
> > Authors

---

### Official Review · Reviewer_4arZ · 2024-11-03

**Soundness:** 2
**Presentation:** 3
**Contribution:** 2
**Rating:** 5
**Confidence:** 5

**Summary:**

The paper presents orthogonal low-rank embedding for self-supervised learning (SSOLE) by decoupling low/high-rank enforcement on positive/negative pairs and low-rank enforcement via deviation matrices.

**Strengths:**

1.The paper is with a good clarity by providing a deep analysis on the problem when applied OLE in SSL. The two challenges to be solved are well discussed.
2.The authors provided a detailed theoretical analysis to illustrate the research problem as well as the developed method.
3.Sufficient experiments are performed, and the results demonstrate the work’s effectiveness.

**Weaknesses:**

1.The main idea of the paper of employing low/high-rank enforcement to adjust the distance between contrastive sample pairs has been proposed and discussed in several previous works of supervised learning (like LDA). As a result, the paper makes incremental contribution by extending this idea to self-supervised learning, which makes its novelty somehow limited. Besides, some related works are not discussed in this paper.
2.The relationship between the three limitations, the two challenges, and the proposed method could be further discussed.
3.The authors consider decoupling the low-rank enforcement and high-rank enforcement with Eq. (2), which needs more explanation to analyze how Eq. (2) achieves this aim.
4.It is mentioned that achieves competitive performance across SSL benchmarks without relying on large batch sizes, memory banks, or dual-encoder architectures, which lacks detailed verification or discussion.

**Questions:**

1.What is the relationship between the three limitations, the two challenges, and the proposed method?

2.How does Eq. (2) decouple the low-rank enforcement and high-rank enforcement?

3.The title of the paper does not mention “multi-view learning”, but why do the authors discuss “multi-view learning” throughout the paper (especially in experiments)? Besides, since there are several multi-view self-supervised learning methods, why not compare the proposed with these works as well?

---

> ### Author Response · Authors · 2024-11-22
> **Response to Reviewer 4arZ (Part 1)**
>
> We are grateful to Reviewer 4arZ for the thoughtful feedback and for recognizing the clarity of our problem formulation, theoretical analysis, and experimental results. Below, we address the concerns in detail, providing clarifications and additional context to emphasize the novelty, rigor, and contributions of our work. We managed to incorporate most of these responses in the revised manuscript in the short time for the revision; what is not yet there will be addressed in the full revision before the publication if being accepted.
>
> # 1. Contribution and Related Work
> >The idea of employing low/high-rank enforcement to adjust the distance between contrastive sample pairs has been explored in supervised learning, making the contribution incremental. Additionally, some related works are not discussed.
>
> We acknowledge the reviewer’s concern and would like to clarify the novelty and distinctiveness of our contribution to SSL:
> - ## Adaptation to SSL is non-trivial:
>   - Supervised methods like Linear Discriminant Analysis (LDA) and OLE benefit from explicit class labels to enforce orthogonality among classes and minimize intra-class rank. However, SSL lacks such kind of supervision, as each instance effectively acts as its own class. This leads to two unique challenges we need to address:
>     - **Infinite “Classes” in SSL**: Orthogonality is impossible to achieve across infinite “classes”, resulting in representational collapse if OLE is naively applied to SSL.
>     - **Misalignment between Positive Pairs**: The nuclear norm fails to distinguish aligned and anti-aligned (opposite direction) vectors, undermining positive pair alignment in SSL.
>    - These challenges are absent in supervised learning and required novel solutions, which are not addressed in previous works. We propose:
>      - **Decoupling low-rank and high-rank enforcement** to independently optimize alignment (positive pairs) and uniformity (negative pairs), and aligning low-rank and high-rank enforcement via nuclear norm optimization with contrastive objectives.
>      - **Introducing deviation matrices** to address the sign ambiguity of cosine similarities, ensuring proper alignment in feature space.
> - ## Novel Metric Integration:
>   Unlike previous works that use low-rank constraints as regularization terms (e.g., LORAC), our approach integrates low-rank and high-rank metrics directly into the SSL loss. This integration provides a principled framework for aligning and separating features, leveraging the full potential of OLE in SSL.
> - ## Related Works:
>   Thanks for your suggestion! We discussed several works in SSL that employ the low/high rank concepts in the Introduction. In the revision, we have expanded our discussion in Section 5.2 to include works like LDA and other supervised methods that employ low/high-rank regularizations. While these works inspire our approach, they do not address the specific challenges of SSL. Our novelty lies in identifying the challenges of extending these ideas to SSL and addressing them through carefully crafted solutions for its unique constraints.

---

> > ### Author Response · Authors · 2024-11-22
> > **Response to Reviewer 4arZ (Part 2)**
> >
> > # 2. Relationship Between Limitations, Challenges, and Method
> > >The relationship between the three limitations, two challenges, and the proposed method could be further discussed.
> >
> > Thank you for pointing out this potential area for improvement. We acknowledge that the mention of "three limitations" appears to be a misunderstanding. In our work, we focus on **two limitations** of the original OLE framework, which are manageable in supervised settings but become problematic in SSL. These limitations, when transferred to SSL, result in two fundamental challenges. Below, we clarify the relationship between these **limitations**, the **challenges** they cause in SSL, and the **proposed method**:
> >
> > - ## Limitations of the Original OLE Framework
> >   - **Highly Entangled Low/High-Rank Enforcement**: In OLE, the low-rank (intra-class similarity) and high-rank (inter-class dissimilarity) objectives are deeply intertwined, making it difficult to independently optimize positive and negative pairs.
> >   - **Nuclear Norm’s Insensitivity to Cosine Similarity Signs**: The nuclear norm used in OLE cannot distinguish between positively and negatively correlated vectors. This limitation is acceptable in supervised settings where labels prevent misalignment, but it becomes critical in SSL due to the absence of supervision.
> >
> > - ## Challenges in Applying OLE to SSL
> >   - **Infinite “Classes” in SSL**: Without labels, SSL treats each instance as its own "class," leading to a virtually infinite number of "classes." This makes it mathematically impossible to enforce inter-class orthogonality, causing the original OLE objective to collapse representations.
> >   - **Misalignment between Positive Pairs**: In SSL, the nuclear norm’s insensitivity to cosine similarity signs leads to the misalignment of positive pairs. This is because anti-aligned (opposite direction) vectors may still contribute to a low-rank representation, undermining the quality of learned features.
> >
> > - ## The Proposed Method
> >   To adapt OLE to SSL, our proposed method, SSOLE, introduces two key strategies:
> >   - **Decoupling Low-Rank and High-Rank Enforcement (Section 4.1)**:
> >     - To address **Challenge 1 (Infinite “Classes” in SSL)**, we decouple low-rank and high-rank enforcement, and align low-rank and high-rank enforcement via nuclear norm optimization with contrastive objectives.
> >     - This disentanglement ensures that the optimization of positive and negative pairs does not interfere with each other, preventing representational collapse.
> >   - **Enhanced Low-Rank Enforcement Using Deviation Matrices (Section 4.2)**:
> >     - To address **Challenge 2 (Misalignment between Positive Pairs)**, we enforce low-rank constraints on deviation matrices rather than directly on the original feature matrices.
> >     - By centering feature vectors (subtracting the mean vector), the deviation matrix focuses on deviations from the mean, ensuring alignment of positive pairs while circumventing the nuclear norm’s insensitivity to cosine similarity signs.
> >
> > - ## Clarifying the Relationship Between Limitations, Challenges, and Method:
> >   - The **limitations** of OLE (entanglement of low/high-rank enforcement and insensitivity to cosine similarity signs) lead to **challenges** in SSL (infinite classes and misalignment of positive pairs).
> >   - These **challenges** are specific to SSL due to the lack of class labels, which prevents the original OLE from working effectively.
> >   - Our **method** (SSOLE) directly addresses these challenges through decoupled enforcement and the use of deviation matrices, as detailed in Section 4.
> >
> > To improve clarity for future readers, we have revised Section 3 to explicitly connect the limitations of OLE to the challenges they create in SSL, and further link these challenges to the solutions described in Section 4. With the additional time after the rebuttal period ends, we will continue revising the paper to further clarify these points.

---

> ### Author Response · Authors · 2024-11-22
> **Response to Reviewer 4arZ (Part 3)**
>
> # 3. Explanation of Equation (2)
> >How does Equation (2) decouple the low-rank enforcement and high-rank enforcement?
>
> Equation (2) introduces separate terms for low-rank (intra-class) and high-rank (inter-class) enforcement, represented by $h_1(\cdot)$ and $h_2(\cdot)$, respectively. This separation enables independent optimization of alignment and uniformity objectives with nuclear norms:
> - **Low-rank Term** $h_1(\cdot)$:
>   - Enforces alignment of positive pairs by minimizing the nuclear norm of the (deviation) matrix ($\tilde{Z}_{b,:}​$).
>   - This ensures that representations of different views of the same instance are aligned.
> - **High-rank Term** $h_2(\cdot)$:
>   - Promotes uniformity by maximizing the nuclear norm of inter-class representations ($Z_{:,n}$).
>   - This ensures that representations from different instances are sufficiently distinct.
>   - By decoupling these terms, Equation (2) avoids the entanglement of alignment and uniformity objectives, which is a key limitation of naive OLE application in SSL.
>
> This is also discussed in Section 4.1, lines 201-214.
>
> # 4. Verification of Computational Claims
> >Claims about computational efficiency lack detailed verification or discussion.
>
> Experiments in Section 6.3 on linear probing, semi-supervised learning, transfer learning show better or competitive performance of SSOLE against other methods. And this is achieved without heavy computational cost. Key points include:
> - **Batch Size**:
>   SSOLE achieves state-of-the-art performance on ImageNet-1k with a batch size of 256, compared to 4096 in BYOL and SimCLR, and 2048 in Barlow Twins and VICReg.
> - **Encoder Architecture**:
>   SSOLE employs a single-branch encoder, unlike dual-encoder architectures in MoCo, BYOL, and LORAC, which use EMA teachers or stop-gradient techniques.
> - **Memory Bank Usage**:
>   Unlike MoCo and LORAC, which rely on large memory banks, SSOLE avoids additional memory requirements.
>
> The claims on computational efficiency are discussed and revised in Section 6.2.
>
> # 5. Multi-view Learning Context
> >The title does not mention multi-view learning, yet the paper discusses it extensively. Why not compare with other multi-view SSL methods?
>
> SSOLE works specially with multi-view learning by leveraging multiple augmentations (views) and optimizing nuclear norms of matrices. We emphasize that our focus is not solely on MV-SSL but on addressing OLE’s challenges within the broader SSL paradigm. In Tables 3 and 4, we compare SSOLE with LORAC and other multi-view methods like SwAV and EMP-SSL. The results show that SSOLE is superior to these works.
>
> We have revised Section 5.1 to show the connection of MV-SSL and LORAC with our work, as well as the differences.
>
> We hope this response addresses the reviewer’s concerns comprehensively and demonstrates the improvements we have made to strengthen the paper. Thank you for your constructive feedback.

---

### Comment · Reviewer_4arZ · 2024-11-26
**about novelty**

The key part of the article, which is the formula for minimizing intra class rank and maximizing inter class rank, already existed at the top conference such as CVPR about six years ago. In addition, the article did not cite the corresponding papers, so the innovation of the article is insufficient. I will keep my score unchanged

---

> ### Author Response · Authors · 2024-11-26
>
> We thank you again for your feedback. Below, we provide a detailed response to the reviewer’s concerns.
> >The key part of the article, which is the formula for minimizing intra class rank and maximizing inter class rank, already existed at the top conference such as CVPR about six years ago.
>
> We respectfully disagree with the reviewer’s assessment of the novelty of our work.
> Our novelty lies in identifying the challenges of extending OLE to SSL and addressing them through carefully crafted solutions for its unique limitations.
>
> As detailed in the paper and our extensive response, while the OLE formulation was presented at CVPR,
> its form is not appropriate and was not developed or used for the important SSL problem,
> resulting in the need for the major changes and improvements introduced in this work.
> We will further stress this again in the revised version, and the companion code will be released with the paper clearly demonstrating it as well.
>
> >In addition, the article did not cite the corresponding papers, so the innovation of the article is insufficient.
>
> While our paper already cites key OLE-related works and highlights their limitations in SSL, we appreciate the reviewer’s feedback regarding additional references.
> We have now included a comprehensive discussion of prior OLE-related works (CVPR and others) in the revised version and will ensure any additional missing references are cited.
>
> Importantly, our paper explicitly acknowledges the roots of OLE in supervised learning while clearly stating its limitations and the novel contributions required to adapt it to SSL.

---

> ### Author Response · Authors · 2024-11-28
> **Follow-Up Comment to Reviewer 4arZ**
>
> Thank you again for your feedback on our submission. We deeply appreciate the effort you have put into evaluating our work. In light of your concerns, we made significant revisions and provided a detailed response.
> Below is a summary of the changes and clarifications we made regarding your concerns in the revised version:
>
> # Unique Contributions:
> While OLE-related ideas have been previously explored in supervised learning, we emphasized that adapting OLE to SSL involves addressing novel and challenging issues, specifically:
>
> - **Infinite “Classes” in SSL**: Unlike supervised settings with labeled classes, SSL treats each instance as its own "class," making it mathematically impossible to achieve inter-class orthogonality with OLE.
> - **Misalignment between Positive Pairs**: The nuclear norm fails to distinguish between aligned and anti-aligned vectors, which undermines the alignment of positive pairs.
>
> To address these, we propose to (1) decouple low-rank and high-rank enforcements and align them with SSL/contrastive objectives; and (2) apply low-rank enforcement on the deviation matrix,
> which are novel solutions specifically designed for SSL and are distinct from earlier OLE applications in supervised settings.
>
> # Relationship Between Limitations, Challenges, and Method:
> We expanded Section 3 to explicitly connect the limitations of OLE in supervised settings to the challenges they cause in SSL and linked these challenges to the solutions presented in Section 4.
> This structure ensures a clear narrative and demonstrates how our contributions address the gaps in applying OLE to SSL.
>
> # Additional Related Work:
> In response to your suggestion, we carefully reviewed relevant works and added discussions on OLE-related methods and other approaches that use low/high-rank regularizations in Section 5.2.
> We also clarified how our work is distinct in adapting these ideas to SSL.
>
> We kindly ask you to review the revised manuscript and our response to better understand the contributions of our work.
> If you believe there are additional papers we should reference, we would greatly appreciate it if you could list them for inclusion in future revisions.
> We are committed to ensuring our work is both rigorous and properly contextualized within the broader research landscape.
>
> Thank you again for your thoughtful feedback and for taking the time to engage with our work.
>
> Best regards,
>
> Anthors

---

### Meta-Review · Area_Chair_iySh · 2024-12-14

**Metareview:**

This paper makes new progress into a rather old framework, named Orthogonal Low-rank Embedding (OLE), receiving positive responses from the majority of the reviewers. I would recommend accepting the paper.

**Additional Comments On Reviewer Discussion:**

After author-reviewer discussions, Reviewer 4arZ still complains about the novelty of the paper: “The key part of the article, which is the formula for minimizing intra class rank and maximizing inter class rank, already existed at the top conference such as CVPR about six years ago. In addition, the article did not cite the corresponding papers, so the innovation of the article is insufficient”.

While it is true that the formula for minimizing intra class rank and maximizing inter class rank is rather old, I agree with the authors that their work makes solid contributions to the community of self-supervised representation learning.

---

### Decision · Program_Chairs · 2025-01-22

Accept (Poster)